# Learning Flexible Generalization in Video Quality Assessment by Bringing Device and Viewing Condition Distributions

**Nikolay Safonov** [1 2]   **Dmitriy Vatolin** [1]

## Abstract

Video quality assessment (VQA) plays a critical role in optimizing video delivery systems. While numerous objective metrics have been proposed to approximate human perception, the perceived quality strongly depends on viewing conditions and display characteristics. Factors such as ambient lighting, display brightness, and resolution significantly influence the visibility of distortions. In this work, we address the question of the multi-screen quality assessment on mobile devices, as this area still tends to be undercovered. We introduce a first large-scale subjective dataset collected across more than different 300 Android devices, accompanied by metadata on viewing conditions and display properties. We propose a strategy for aggregated score extraction and adaptation of VQA models to device-specific quality estimation. Our results demonstrate that incorporating device and context information enables more accurate and flexible quality prediction, offering new opportunities for fine-grained optimization in streaming services. Ultimately, this work advances the development of perceptual quality models that bridge the gap between laboratory evaluations and the diverse conditions of real-world media consumption. We made the dataset and the code available at https://videoprocessing.github.io/device-viewing-conditions.

## 1. Introduction

Objective video quality assessment (VQA) is a critical task in video processing, with applications in compression,

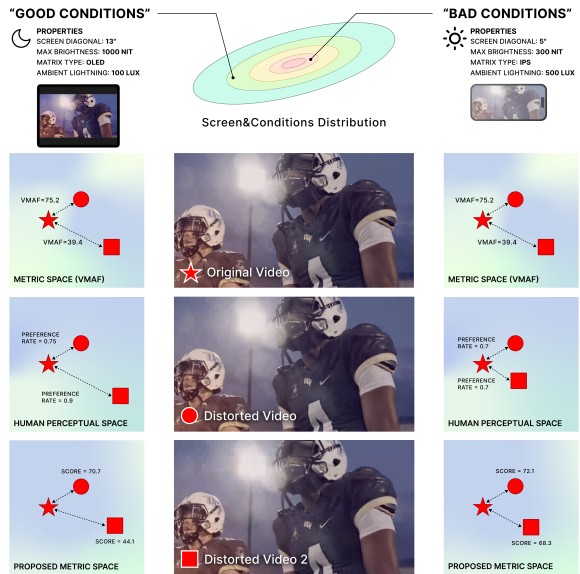

*Figure 1.* Depiction of the multi-screen video quality assessment task problem. The perceived quality strongly depends on viewing conditions and display characteristics. Videos with visibly different quality on the high-end devices may be perceived with the same quality on the low-end ones, and existing VQA metrics do not take that into account.

streaming, and content delivery. The goal of VQA models is to estimate human-perceived video quality using algorithmic predictions. In this work we focus on compressed video quality assessment. To train and evaluate these models, researchers rely on benchmark datasets that contain subjective human scores, typically collected in controlled lab environments or through crowd-sourced studies (Wang et al., 2016a; 2017; Zhu et al., 2016; Barman et al., 2018; Antsiferova et al., 2022; Rao et al., 2024).

A key challenge is that human perception of video quality is not invariant, it varies significantly depending on viewing conditions and display characteristics, such as screen size, resolution, brightness, and ambient lighting. For example, in (Barman et al., 2023), a parallel subjective test was conducted on a phone, tablet, and television. The results showed substantial differences in Mean Opinion Scores (MOS) obtained across the different devices. However, most existing

[1]MSU Institute for Artificial Intelligence, Moscow, Russia [2]AI Center, Lomonosov Moscow State University, Moscow, Russia. Correspondence to: Nikolay Safonov <nikolay.safonov@graphics.cs.msu.ru>.

*Proceedings of the 43rd International Conference on Machine Learning*, Seoul, South Korea. PMLR 306, 2026. Copyright 2026 by the author(s).

datasets and VQA models either neglect these contextual factors or assume a uniform viewing environment. As a result, objective metrics that perform well in one setting may fail to generalize across diverse real-world conditions. For example, as shown in the example on Figure 1, videos with a visibly different quality on high-end devices may be perceived with the same quality on low-end ones. This issue is particularly pronounced in the case of banding metrics, since banding artifacts appear differently depending on screen brightness. As shown in (Safonov et al., 2024), current banding metrics demonstrate very low reliability. For the same reason, the developers of the state-of-the-art Netflix VMAF metric provide different models separately for TVs and mobile phones.

While numerous VQA models (Zhang et al., 2018; Ding et al., 2020; Li et al., 2016; Wu et al., 2023) have been proposed in recent years, they typically evaluate performance on datasets with limited variability in device types and viewing environments. This limits their applicability to real-world use cases, particularly for mobile users and diverse consumer devices. While mobile devices often share similar characteristics in terms of screen size and resolution, the actual viewing conditions for mobile users can vary significantly. For instance, watching the same video scene outdoors in bright sunlight may result in a drastically different perceptual experience compared to viewing it in a dark room. Additionally, many users deliberately reduce screen brightness to conserve battery life, further affecting visual quality.

In recent years, a noticeable gap has emerged between high dynamic range (HDR) capable and standard dynamic range (SDR) only mobile devices. Modern flagship smartphones are equipped with HDR displays capable of peak brightness levels exceeding 2500 nits, offering enhanced visibility in bright environments and more accurate rendering of dark scenes. A detailed analysis of how SDR and HDR content is perceived on HDR-enabled displays is provided in (Ebenezer et al., 2024), highlighting additional complexities introduced by display capabilities when assessing visual quality. Without accounting for such contextual differences, VQA models risk producing misleading predictions, which can hinder user experience optimization for content providers and device manufacturers.

In this work, we address the problem of multi-screen quality assessment. First off all we collect a large-scale, crowd-sourced multi-screen video quality dataset designed to bridge this realism gap. The dataset comprises pairwise preference judgments on 300+ unique Android devices, enriched with detailed metadata: device model, screen technology, diagonal size, peak brightness, applied brightness setting, measured ambient light etc. The dataset also includes pairwise comparison judgments and the "reference" scores collected on high-resolution desktop monitors.

On this dataset, we evaluate existing learning-based image quality assessment (IQA) and VQA models and show their limited ability to preserve correct quality orderings across different viewing conditions. To address this, we propose a training strategy and a vote aggregation framework that generalize VQA metrics for improved performance under diverse device-specific conditions. To the best of our knowledge, this is the first framework that explicitly trains quality metrics to account for viewing-device characteristics. Our findings show that incorporating device and context information substantially improves prediction accuracy and robustness across viewing conditions that can serve as a foundation for new generations of adaptive streaming solutions. Our contributions are as follows:

- We introduce a large-scale dataset with over 250K pairwise comparisons collected on 300+ mobile devices, enriched with detailed viewing-condition metadata.

- We propose a practical adaptation algorithm to fit objective quality metrics predictions for the different viewing conditions;

- We propose using a viewing conditions pool, which enables training models on conditions not explicitly represented in the subjective dataset, thereby improving robustness and generalization performance.

## 2. Related works

The diversity of modern display mobile devices has important implications across many areas of video processing. However, in this work, we focus specifically on video compression, which remains one of the most fundamental and widespread applications where perceptual video quality plays a critical role. Another broad category of subjective datasets involves in-the-wild distortions, typically captured by non-professionals and characterized by artifacts such as shaking, blurring, and faded colors. These distortions are primarily related to the aesthetic aspects of content and may require separate quality assessment approaches, as shown in (Wu et al., 2023). Notably, the perceived quality in such cases tends to remain relatively consistent across different display screens. Consequently, we concentrate our comparisons on compression-oriented and general quality video quality datasets and benchmarks. A summary of relevant datasets is provided in Table 1.

Numerous datasets have been proposed to support the development and evaluation of perceptual video quality metrics, particularly in the context of video compression and streaming. Large-scale just noticeable difference (JND) based datasets such as VideoSet (Wang et al., 2016b) and MCL-JCV (Wang et al., 2016a) enable fine-grained analy-

*Table 1.* Summary of subjective compressed video quality datasets including multi-screen and the proposed dataset.

| | Dataset | Content | Orig. | Dist. | Device number | Subjective Framework | Subj. | Ans. |
|---|---|---|---|---|---|---|---|---|
| **General** | MCL-JCV (2016) (Wang et al., 2016a) | SDR | 30 | 1,560 | - | In-lab | 150 | 78K |
| | VideoSet (2017) (Wang et al., 2017) | SDR | 220 | 45,760 | - | In-lab | 800 | - |
| | SJTU-4K (2017) (Zhu et al., 2016) | SDR | 20 | 200 | - | In-lab | 30 | 6K |
| | GamingVSET (2018) (Barman et al., 2018) | SDR | 24 | 576 | - | In-lab | 25 | - |
| | NFLX (2016) (Li et al., 2016) | SDR | 12 | 300 | - | In-lab | 54 | 9K |
| | KUGVD (2019) (Barman et al., 2019) | SDR | 6 | 144 | - | In-lab | 17 | - |
| | UGC-VIDEO (2020) (Li et al., 2020b) | SDR | 50 | 550 | - | In-lab | 30 | 16.5K |
| | AVT-VQDB (2019) (Rao et al., 2019) | SDR | 15 | 300 | - | In-lab | 50 | 15K |
| | TGV (2022) (Wen et al., 2022) | SDR | 150 | 1,143 | - | In-lab | 19 | - |
| | TaoLive (2023) (Zhang et al., 2023) | SDR | 418 | 3,762 | - | In-lab | 44 | 165.5K |
| | KVQ (2024) (Lu et al., 2024) | SDR | 600 | 4,200 | - | In-lab | 15 | 63K |
| | CVQAD (2022) (Antsiferova et al., 2022) | SDR | 36 | 1,022 | - | Crowd. | 10,800 | 320K |
| | LEHA-CVQAD (2025) (Gushchin et al., 2025) | SDR | 59 | 6,240 | - | Crowd. | 11,000 | 400K |
| | YT-UGC+ (2021) (Wang et al., 2021) | SDR/HDR | 189 | 567 | - | In-lab | 30 | 17K |
| | HDR-sport (2023) (Shang et al., 2023) | HDR | 12 | 42 | - | In-lab | 140 | 32K |
| | BrightVQA (2025) (Authors, 2025) | HDR | 300 | 2100 | - | Crowd. | 200 | 74K |
| | AVT-VQDB-UHD-1-HDR(2024) (Rao et al., 2024) | HDR | 5 | 195 | - | In-lab | 24 | 4,7K |
| | Shang2022 (2022) (Shang et al., 2022) | HDR | 31 | 310 | - | In-lab | 66 | 22K |
| **Multi-screen** | MSVSA (2023) (Barman et al., 2023) | SDR | 4 | 36 | 3 | In-lab | 26 | - |
| | MS-Banding (2024) (Safonov et al., 2024) | SDR | 15 | 120 | 3 | In-lab | 186 | 9K |
| | HDRSDR-VQA (2025) (Chen et al., 2025a) | SDR/HDR | 54 | 960 | 6 | In-lab | 145 | 22K |
| | HDRorSDR (2024) (Ebenezer et al., 2024) | SDR/HDR | 25 | 356 | 3 | In-lab | 67 | 23K |
| | **Proposed** | SDR/HDR | 25 | 650 | 300+ | Crowd. | 10,000 | 240K |

sis of compression artifacts, while classic datasets like the H.264/AVC video database (Nuutinen et al., 2016) provide foundational resources for benchmarking quality metrics. Recent efforts have also introduced compression-oriented benchmarks tailored to learning-based approaches, as seen in the Video Compression Dataset and Benchmark (Antsiferova et al., 2022). To address streaming-specific challenges, MCL-V (Lin et al., 2015) simulates bitrate fluctuations and stalling, and GamingVideoSet (Barman et al., 2018) along with related machine learning approaches (Barman et al., 2019) focus on passive gaming video quality estimation. Similarly, user-generated content (UGC) and its perceptual variability are examined in UGC-Video (Li et al., 2020b), which highlights the aesthetic and artifact-rich nature of such content. Other works investigate factors influencing subjective perception beyond compression, temporal effects on video quality of experience (Bampis et al., 2017), and Quality of Service parameters (Fiedler et al., 2010). High-quality reference datasets such as the TUM HD Video Datasets (Keimel et al., 2012) provide additional controlled material for model development and evaluation. Recent works (Shang et al., 2023; Authors, 2025; Rao et al., 2024) have also introduced datasets containing compressed HDR video.

Several studies explicitly investigate the impact of viewing conditions by modeling them under controlled laboratory settings. For example, experiments conducted under varying ambient illumination conditions have been proposed to analyze their influence on perceived visual quality (Shang et al., 2022). Other work introduce test protocols in which both SDR and HDR content are evaluated across different display devices employing diverse panel technologies (Ebenezer et al., 2024). Additionally, the effect of viewing distance on perceived visual quality has been systematically studied (Keller et al., 2024).

Several metrics that model the human visual system (HVS) have been developed to explicitly account for different viewing conditions; for example, in (Mantiuk et al., 2024; 2023; Narwaria et al., 2015), the authors model display type and viewing distance by adjusting the parameters of HVS perceptual response curves. Models to of operate consistently on both SDR and HDR compressed video have been developed (Chen et al., 2025b).

In (Barman et al., 2023), a small-scale dataset was collected using three display devices: tablet, phone, and TV viewed in parallel. The dataset includes Mean Opinion Scores (MOS) for each device type, revealing notable differences in perceived quality across screens. In (Chen et al., 2025a) authors note that visual quality may vary across different viewing conditions and therefore conduct subjective testing on six devices; however, the reported results are aggregated into the single scores, without an analysis of multi-screen effects. In (Safonov et al., 2024), the authors investigate the performance of banding metrics across three domains: TV, MacBook, and crowd-sourcing platforms. To support this analysis, they introduce a dataset comprising MOS scores for 120 distorted video variants. However, the dataset in (Safonov et al., 2024) is distortion-specific, with a strong emphasis on videos containing flat regions, which are particularly susceptible to banding artifacts.

Existing datasets, regardless of distortion type, typically

assume controlled viewing conditions or focus on a single device type. As such, they are not well-suited for exploring the impact of device diversity and ambient conditions on perceived quality. Our work fills this gap by introducing a large-scale, multi-device dataset with associated viewing metadata, allowing for more representative evaluation of compression quality under realistic usage scenarios.

## 3. Dataset

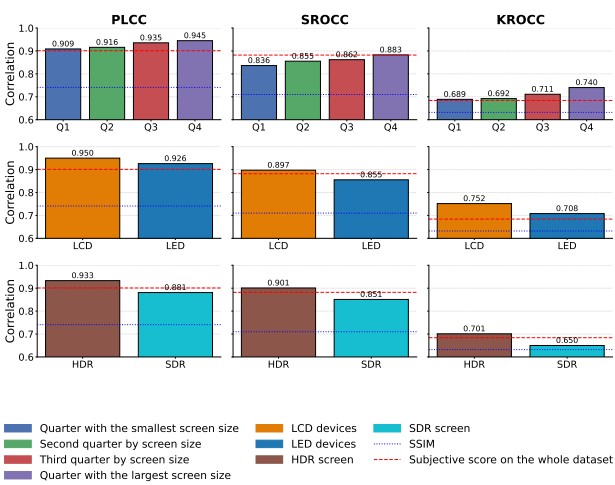

*Figure 2.* **Top row**: Bradley–Terry (Bradley & Terry, 1952) aggregated scores for subsets of devices grouped by screen size (the smallest quarter, the two middle quarters, and the largest quarter) and comparing them with aggregated scores obtained on desktop devices with large screens by Pearson (PLCC), Spearman (SROCC) and Kendall (KROCC) correlations;
**Middle row**: correlations separately for LED and LCD displays;
**Bottom row**: correlations on HDR videos separately for HDR-capable and SDR displays.

### 3.1. Dataset construction

We constructed a compression dedicated dataset exhibiting diverse compression artifacts. For reliable evaluation, reference videos must be of sufficiently high quality to avoid confounding recompression effects. We therefore sampled from over 18,000 high-bitrate open-source videos available on Vimeo under appropriate licenses. Only videos with a minimum bitrate of 20 Mbps were retained, resulting in a collection with an average bitrate of 130 Mbps.

To ensure representative coverage of spatial and temporal complexities, we performed clustering in the complexity space. Spatial complexity was measured as the average size of x264-encoded I-frames normalized by uncompressed frame size, while temporal complexity was defined as the ratio of average P-frame to I-frame sizes. The source videos were selected via clustering in the spatial–temporal complexity space. We introduced compression artifacts by en-

coding them with five encoders spanning different standards (HEVC, VVC, and AV1). The details on encoding settings and source video properties may be found in Appendix B.1.

### 3.2. Subjective testing

To collect pairwise preference annotations across a wide range of devices and viewing conditions, we relied on crowd-sourcing. To ensure precise control over device characteristics and viewing environments, we developed an Android application that automatically recorded the device model, screen specifications, and contextual information such as brightness, ambient luminance, and orientation. Participants recruited via a crowd-sourcing platform were asked to install the application, which both launched the video quality assessment interface and continuously logged relevant metadata. Screen brightness and ambient light levels were sampled every second using system APIs and the device's light sensor, when available. By default, participants used their own brightness settings; however, the application also supported enforced brightness levels. Using this feature, we collected additional subsets with brightness fixed to maximum and minimum values. The details on the collected characteristics, metadata, filtering and parameters distributions may be found in Appendix B.3.

Our subjective study followed a pairwise comparison protocol, where for each source video we generated all possible pairs of its compressed versions. The reference source video itself was also included in the pool. Participants watched the pairs sequentially in full-screen mode and indicated which video exhibited better visual quality, or selected an "equal quality" option. They were allowed to replay the videos before making a choice. Each participant completed 12 comparisons, two of which were control pairs with an obvious quality difference; responses from participants who failed these checks were discarded. To improve the robustness of the results, a minimum of 15 judgments was collected for every pair. In total, the study yielded 250,000 valid annotations from nearly 10,000 contributors. Dataset parameters are summarized in Table 1.

We further enriched the dataset with HDR video content. For these sequences, we used high-quality reference subjective labels from (Rao et al., 2024). During crowd-sourced labeling, we did not require participants to use HDR-capable displays; therefore, on SDR devices the HDR videos were viewed in automatically tone-mapped form provided by the playback pipeline.

## 4. Blade-Chest model

Different models can be applied to aggregate pairwise preference votes. The most commonly used approach is the Bradley–Terry (Bradley & Terry, 1952) model, which has

been employed in large-scale datasets such as (Antsiferova et al., 2022) and (Gushchin et al., 2025). An alternative is the Elo rating system (Elo, 1978), which estimates latent scores through iterative updates after each comparison. However, such models do not account for the viewing conditions under which comparisons are made, even though these conditions can significantly influence the results. Figure 2 illustrates this effect by showing Bradley–Terry (Bradley & Terry, 1952) aggregated scores for subsets of devices grouped by screen size (the smallest quarter, the two middle quarters, and the largest quarter) and comparing them with aggregated scores obtained on desktop devices with large screens by Pearson (PLCC), Spearman (SROCC) and Kendall (KROCC) correlations. The correlations steadily increase from the smallest to the largest screen groups. Figure 2 also reports correlations separately for LED and LCD displays. It may be noted that LCD displays correlate with the desktop scores much better than LED displays. This could be explained by the fact that most desktop monitors also use LCD matrices, which, for example, perform worse in rendering dark colors. Also Figure 2 presents a comparison of HDR and SDR displays for HDR video quality assessment. While evaluations performed on HDR-capable screens show higher correlation with the reference subjective scores, the difference compared to assessments conducted on SDR displays is not radical. This observation is consistent with previously reported results (Ebenezer et al., 2024).

The Blade-Chest model (Chen & Joachims, 2016) leverages this limitations and makes possible to encounter conditions under which each pair were compared, as it proposes learning two vectors for each player $q_i$, namely $q_i^{\text{blade}}$ and $q_i^{\text{chest}}$. The probability of $q_i$ defeating $x_j$ is then determined by comparing the distances between these representations: if $q_i^{\text{blade}}$ is closer to $q_j^{\text{chest}}$ than $q_j^{\text{blade}}$ is to $q_i^{\text{chest}}$, player $q_i$ is predicted to win. The use of the "blade" and "chest" embeddings provides an intuitive interpretation of the underlying model.

Therefore, we adopt the Blade–Chest model (Chen & Joachims, 2016) for our task, as it naturally incorporates information about the viewing conditions into the score aggregation process. To obtain rank scores from pairwise votes, we assumed that the probability of video $i$ being preferred over video $j$ in the viewing conditions $z$ is given by:

$$P(i \succ j, z) = \sigma(||f_c(q_i, z) - f_b(q_j, z)||^2 - \\ ||f_c(q_j, z) - f_b(q_i, z)||^2) \quad (1)$$

where $q_i$ and $q_j$ are the desired subjective score estimates of videos $i$ and $j$, $\sigma(\cdot)$ denotes the sigmoid function, while $f_c(\cdot)$ and $f_b(\cdot)$ are transformation functions conditioned on the subjective score estimates and viewing conditions.

These functions output $q_i^{\text{chest}}$ and $q_j^{\text{blade}}$, respectively. The functions $f_c(\cdot)$ and $f_b(\cdot)$ can take different forms; however, in this work we initialize them as fully connected neural networks, parameterized by $\theta$ and $\psi$, respectively. For the activation functions we use $\tanh$, in order to avoid purely linear behavior. Our experiments have shown that, due to the dominant influence of $q_i$ over $z$, the networks tend to degenerate into linear mappings without nonlinear activations. The vector $z$ was defined as a five-dimensional representation containing information about the display's physical size, pixel resolution, brightness, surrounding luminance, and display type.

To estimate the latent values $q_i$ and the network parameters $\theta$ and $\psi$, we employed a two-step optimization procedure based on the expectation–maximization approach. In our formulation, the latent subjective quality scores $q_i$ cannot be directly observed, while the available supervision comes only from the pairwise preference votes $\mathcal{D} = \{(i, j, z)\}$. We therefore treat $q_i$ as latent variables and optimize them jointly with the network parameters $\theta$ and $\psi$ via the Expectation–Maximisation (EM) algorithm.

The goal of the EM procedure is to obtain a stable conditional decomposition that maximizes the pairwise likelihood. Use of EM is motivated by the fact that the latent quality scores $q_i$ are not directly observed, while supervision is available only through pairwise votes under conditions $z$. The supplementary A makes this explicit by defining the complete-data likelihood, the E-step as updating the latent scores under the current network parameters (current conditions), and the M-step as updating the parameters of $f_c$ and $f_b$ through maximization of the surrogate objective.

The complete-data likelihood of observing a pairwise vote $(i \succ j, z)$ can be expressed as:

$$\mathcal{L}_c(q, \theta, \psi \mid \mathcal{D}) = \prod_{(i,j,z) \in \mathcal{D}} P(i \succ j, z \mid q, \theta, \psi). \quad (2)$$

Taking the logarithm, the complete-data log-likelihood becomes:

$$\log \mathcal{L}_c(q, \theta, \psi) = \\ \sum_{(i,j,z) \in \mathcal{D}} \log \sigma \Big( ||f_c(q_i, z; \theta) - f_b(q_j, z; \psi)||^2 - \\ ||f_c(q_j, z; \theta) - f_b(q_i, z; \psi)||^2 \Big). \quad (3)$$

Thus, the EM procedure allows us to jointly infer the latent subjective quality scores $q_i$ and optimise the transformation networks $f_c(\cdot)$ and $f_b(\cdot)$ under varying viewing conditions. A more detailed derivation of the update rules and implementation details of the optimisation procedure are provided in Appendix A. The obtained subjective scores exhibit a high correlation with those derived using the Bradley–Terry

model on both the mobile and desktop datasets, indicating that the learned representation is consistent and logically grounded. Table 2 reports the obtained scores correlations with the reference scores and scores aggregated using Bradley-Terry model.

*Table 2.* Correlation of the Blade-Chest scores with the reference and Bradley-Terry scores.

|  | PLCC | SROCC | KROCC |
|---|---|---|---|
| Desktop scores | $0.898 \pm 0.063$ | $0.864 \pm 0.059$ | $0.770 \pm 0.070$ |
| BT-scores | $0.912 \pm 0.059$ | $0.881 \pm 0.055$ | $0.808 \pm 0.067$ |

## 5. Conditions adaptation

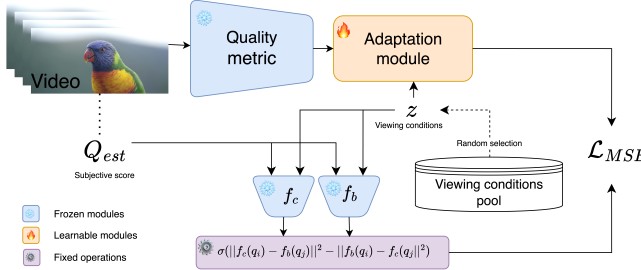

*Figure 3.* The training framework scheme: from the video set a random pair is sampled, and viewing conditions are selected from a distribution. The video quality metric predictions for the selected videos and the viewing conditions are processed through the adaptation module, while the estimated subjective scores together with the viewing conditions are processed through the match function using $f_c(\cdot)$ and $f_b(\cdot)$.

Although we obtained subjective scores $q_i$, these values alone are of limited interest, since it is nontrivial to interpret or compare absolute score levels directly. We also trained fully connected neural networks, $f_c(\cdot)$ and $f_b(\cdot)$, which operate on score pairs; however, this formulation is not well suited for VQA applications, where it is often necessary to estimate the quality of a single video. Therefore, we treat the extraction of subjective scores as an intermediate step in adapting VQA metrics for quality prediction under varying viewing conditions.

The goal of VQA model adaptation is to enable predictions of relevant quality labels under specific viewing conditions. This is particularly useful for streaming platforms, which often optimize compression strategies for certain device types and have access to user distributions and profiles, yet still rely on standard VQA models that may under- or over-estimate the quality perceived by the end users. Modern learning-based models, both deep and traditional (e.g., VMAF), are typically trained on large-scale datasets, so retraining them directly on our proposed dataset may be not sufficient, even when targeting condition-dependent predic-

tion. To address this, we fine-tune the models by passing their predictions to a condition adaptation module.

The adaptation module is a lightweight fully connected neural network trained separately for each VQA model. It takes as input the model prediction together with the target viewing conditions $z$, and outputs a single value representing the predicted video quality under these conditions. The training samples are drawn from the proposed dataset. For training, the viewing conditions $z$ are generated by randomly sampling parameters from a uniform distribution, subject to hand-picked physically motivated constraints.

Now, when we have obtained subjective scores $q_i$ and trained the networks $f_c(\cdot)$ and $f_b(\cdot)$ on the real predictions, we can use their combination to expand the training set with simulated samples. Since the set of videos is fixed and new videos cannot be added without additional human judgments, we instead simulate new viewing conditions. Conditions $z$ are sampled from a uniform distribution and passed to $f_c(\cdot)$ and $f_b(\cdot)$ together with a randomly selected pair of distorted versions of the same source video. This yields the probability that video $i$ is of higher quality than video $j$ under the given conditions $z$.

So the training framework is as follows: from the video set a random pair is sampled, and viewing conditions are selected from a uniform distribution. The video quality metric predictions for the selected videos and the conditions are processed through the adaptation module, while the estimated subjective scores together with the viewing conditions are processed through the match function using $f_c(\cdot)$ and $f_b(\cdot)$. The overall framework is illustrated in Figure 3.

The adaptation network was implemented as a fully connected neural network of depth 4, with hidden layers of size 64. We used tanh activations in the hidden layers to avoid linearity and applied a sigmoid activation at the output to constrain predictions to the $[0, 1]$ range.

We employ a MLP to capture nonlinear dependencies and implicit interactions between display-related parameters and perceived visual quality. Compared to generalized additive models (GAM) and attention-based tabular models such as TabNet (Arik & Pfister, 2021), MLP offers a better trade-off between expressiveness, robustness, and computational efficiency in low-data subjective settings, while remaining interpretable.

The condition pool is a key component motivated by the inherent sparsity of subjective data collected under heterogeneous viewing conditions. In the first stage, the Blade–Chest aggregation operates on sparse pairwise comparisons collected under varying conditions to estimate latent subjective scores and learn condition-aware transformations. Once these scores are obtained, the original sparse observations are no longer sufficient for robust training of downstream

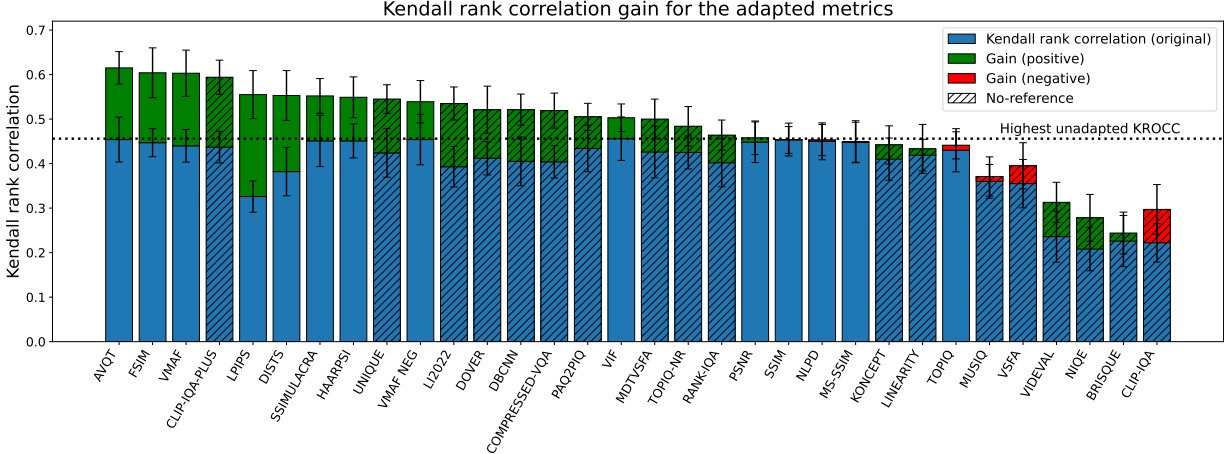

*Figure 4.* Kendall rank correlations for the original metrics and their adapted counterparts. Gain represents how score improved after metric training (green = positive, red = negative).

models. The condition pool addresses this by enabling sampling of viewing conditions independently from the original annotations, effectively augmenting the training process and improving generalization, including to conditions not explicitly observed in the dataset. Importantly, this mechanism is only feasible due to the use of the Blade–Chest model, which explicitly incorporates viewing conditions into the aggregation process.

As a result, the adaptation module is able to predict video quality based on both the metric predictions and the viewing conditions. We trained separate adaptation modules for each of the considered metrics. In addition, we conducted experiments where the adaptation module was used as a fusion mechanism across multiple metrics.

## 6. Evaluation

For evaluating the performance of VQA models, correlations with subjective quality scores are commonly employed. Pearson and Spearman correlations are appropriate when aggregated subjective scores (e.g., MOS, Bradley–Terry estimates) are available. However, in our dataset each crowdsourcing assessor evaluated only a small portion of the content. Consequently, the number of samples collected under identical viewing conditions is insufficient to construct reliable subjective scores for a fixed screen setup. In contrast, Kendall rank correlation relies on the proportion of correctly ordered pairs. Since the subjective annotations in our dataset consist of pairwise preference selections under different measured viewing conditions, Kendall rank correlation provides a natural criterion for assessing whether a VQA metric preserves the quality ordering implied by human judgments. Therefore, we adopt Kendall rank correlation as the primary evaluation metric.

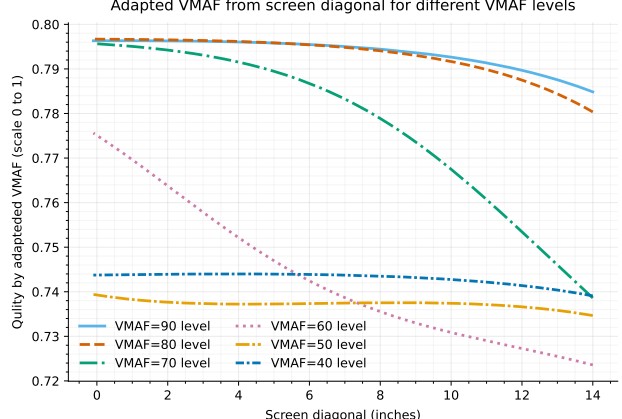

*Figure 5.* Relationship between the estimated quality of the VMAF adaptation module and the viewing conditions, predictions for several fixed VMAF levels with varying the screen diagonal (all other parameters are held constant).

Five source videos, along with all their distorted versions, were held out as the testing set. We first applied both classical and modern neural network–based image and video quality metrics to these videos. For each participant's vote, we derived the predicted ordering from the metric outputs and then computed Kendall rank correlation with the subjective preference. Subsequently, we trained an adaptation fully connected network separately for each of the evaluated metrics on the training portion of the dataset and also tested by Kendall rank. Figure 4 demonstrates the Kendall rank correlations for the original metrics and their adapted counterparts.

It can be observed that even state-of-the-art metrics exhibit relatively low correlations on the raw vote data compared to the aggregated scores. This is expected, as the raw an-

*Table 3.* Kendall rank correlations for the original metrics and their adapted counterparts over three most common phone models. Gain represents how score improved after metric training (red = positive, blue = negative).

| | Metric | Phone model 1 | | | Phone model 2 | | | Phone model 3 | | |
|---|---|---|---|---|---|---|---|---|---|---|
| | | KROCC | KROCC (adapted) | Gain | KROCC | KROCC (adapted) | Gain | KROCC | KROCC (adapted) | Gain |
| Full-reference | LPIPS (Zhang et al., 2018) | 0.308 | 0.652 | +0.344 | 0.404 | 0.548 | +0.145 | 0.313 | 0.631 | +0.318 |
| | DISTS (Ding et al., 2020) | 0.338 | 0.610 | +0.272 | 0.340 | 0.517 | +0.178 | 0.399 | 0.550 | +0.152 |
| | VMAF (Li et al., 2016) | 0.509 | 0.600 | +0.091 | 0.498 | 0.595 | +0.097 | 0.548 | 0.572 | +0.024 |
| | AVQT (Sodhani, 2021) | 0.439 | 0.694 | +0.255 | 0.427 | 0.613 | +0.186 | 0.478 | 0.584 | +0.106 |
| | FSIM (Zhang et al., 2011) | 0.387 | 0.634 | +0.247 | 0.419 | 0.645 | +0.226 | 0.439 | 0.617 | +0.178 |
| | SSIMULACRA (Sneyers, 2023) | 0.457 | 0.635 | +0.178 | 0.480 | 0.541 | +0.061 | 0.412 | 0.634 | +0.222 |
| | HAARPSI (Kastryulin et al., 2019) | 0.481 | 0.526 | +0.045 | 0.454 | 0.552 | +0.098 | 0.436 | 0.510 | +0.074 |
| | VMAF NEG | 0.416 | 0.565 | +0.149 | 0.469 | 0.563 | +0.094 | 0.470 | 0.535 | +0.065 |
| | VIF (Sheikh & Bovik, 2006) | 0.449 | 0.535 | +0.086 | 0.487 | 0.505 | +0.018 | 0.449 | 0.543 | +0.094 |
| | PSNR | 0.469 | 0.478 | +0.009 | 0.448 | 0.491 | +0.043 | 0.501 | 0.430 | -0.071 |
| | MS - SSIM (Wang et al., 2003) | 0.427 | 0.451 | +0.024 | 0.485 | 0.454 | -0.031 | 0.454 | 0.405 | -0.049 |
| | SSIM | 0.415 | 0.511 | +0.096 | 0.481 | 0.500 | +0.019 | 0.456 | 0.459 | +0.003 |
| | NLPD | 0.473 | 0.485 | +0.012 | 0.468 | 0.415 | -0.053 | 0.491 | 0.431 | -0.060 |
| | TOPIQ (Chen et al., 2024) | 0.484 | 0.498 | +0.014 | 0.438 | 0.474 | +0.036 | 0.479 | 0.456 | -0.023 |
| No-reference | CLIP-IQA-PLUS (Wang et al., 2023b) | 0.417 | 0.581 | +0.164 | 0.427 | 0.607 | +0.180 | 0.463 | 0.572 | +0.109 |
| | LI2022 (Li et al., 2022) | 0.434 | 0.539 | +0.105 | 0.403 | 0.468 | +0.065 | 0.398 | 0.566 | +0.168 |
| | UNIQUE (Zhang et al., 2021) | 0.466 | 0.608 | +0.142 | 0.447 | 0.516 | +0.069 | 0.422 | 0.596 | +0.174 |
| | DBCNN (Zhang et al., 2020) | 0.373 | 0.598 | +0.225 | 0.414 | 0.486 | +0.072 | 0.374 | 0.522 | +0.148 |
| | COMPRESSED - VQA (Sun et al., 2021) | 0.351 | 0.568 | +0.217 | 0.361 | 0.551 | +0.190 | 0.423 | 0.496 | +0.073 |
| | DOVER (Wu et al., 2023) | 0.491 | 0.580 | +0.089 | 0.430 | 0.565 | +0.135 | 0.422 | 0.586 | +0.164 |
| | VIDEVAL (Tu et al., 2021) | 0.283 | 0.407 | +0.124 | 0.264 | 0.243 | -0.021 | 0.266 | 0.322 | +0.056 |
| | MDTVSFA (Li et al., 2021) | 0.445 | 0.532 | +0.087 | 0.482 | 0.496 | +0.014 | 0.456 | 0.481 | +0.025 |
| | PAQ2PIQ (Ying et al., 2020) | 0.395 | 0.612 | +0.217 | 0.393 | 0.494 | +0.101 | 0.451 | 0.535 | +0.084 |
| | NIQE | 0.187 | 0.306 | +0.119 | 0.250 | 0.249 | -0.001 | 0.194 | 0.315 | +0.121 |
| | RANK - IQA (Liu et al., 2017) | 0.470 | 0.503 | +0.033 | 0.394 | 0.450 | +0.056 | 0.451 | 0.498 | +0.047 |
| | TOPIQ - NR (Chen et al., 2024) | 0.444 | 0.526 | +0.082 | 0.440 | 0.502 | +0.062 | 0.414 | 0.457 | +0.043 |
| | KONCEPT (Hosu et al., 2020) | 0.376 | 0.444 | +0.068 | 0.421 | 0.422 | +0.001 | 0.454 | 0.438 | -0.016 |
| | BRISQUE (Mittal et al., 2012) | 0.246 | 0.279 | +0.033 | 0.198 | 0.242 | +0.044 | 0.210 | 0.251 | +0.041 |
| | LINEARITY (Li et al., 2020a) | 0.367 | 0.435 | +0.068 | 0.376 | 0.400 | +0.024 | 0.381 | 0.437 | +0.056 |
| | MUSIQ (Chen et al., 2024) | 0.320 | 0.320 | -0.000 | 0.306 | 0.325 | +0.019 | 0.402 | 0.379 | -0.023 |
| | VSFA (Li et al., 2019) | 0.347 | 0.380 | +0.033 | 0.392 | 0.419 | +0.027 | 0.333 | 0.446 | +0.113 |
| | CLIP-IQA (Wang et al., 2023a) | 0.222 | 0.227 | +0.005 | 0.329 | 0.281 | -0.048 | 0.185 | 0.169 | -0.016 |

notations are inherently noisy: for the same video under identical conditions, different participants may prefer different versions. Moreover, individual differences such as prior viewing experience or eye health can further contribute to variability. Nevertheless, despite the use of visual acuity tests and validation procedures on the crowd-sourcing platform, the same participant may occasionally select different options for the same video pair. Another important factor is the viewing condition, which strongly influences perceived quality but is not explicitly modeled by existing metrics. Nevertheless, the adapted versions of the metrics substantially improve performance. While adaptation consistently enhances correlations, the remaining label noise limits performance.

To study the out-of-distribution generalization of the proposed approach, we evaluate the model on the three most frequent smartphone models, each accounting for more than 2% of the collected votes: Xiaomi Redmi Note 13 (model 1), Samsung Galaxy A55 (model 2), and Xiaomi Redmi Note 8 Pro (model 3). During training, all samples associated with each target device were excluded, and evaluation was performed separately on the held-out device. Table 3

shows Kendall correlations of metrics, adapted metrics and the gain. It can be observed that, for most metrics, training consistently improves performance, indicating good generalization. This behavior may be attributed to the use of condition pool, which enables robust learning across different combinations of display parameters, including those not explicitly present in the subjective test data.

Since the adaptation module employs a sigmoid activation function, the predicted scores are scaled between 0 and 1, which makes the model's outputs fairly interpretable. To illustrate the relationship between the estimated quality of the VMAF adaptation module and the viewing conditions, we plot predictions for several fixed VMAF levels while varying only the screen diagonal (all other parameters are held constant). Figure 5 shows this dependency. As expected, the perceived quality decreases as the screen size increases. It should also be noted that beyond the observed range (our dataset includes only a limited number of devices with diagonals larger than 8 inches), the predicted curves become less reliable. For the higher levels of VMAF the screen enlargement is more sufficient, as people tend to star seeing the artifacts, similar effect demonstrated in (Wang et al., 2024).

Also we conducted SHAP (Lundberg & Lee, 2017) analysis of display-related factors influencing perceived visual quality. Display brightness and physical screen size exhibit the strongest contribution to the model predictions. Pixel density and surrounding luminance show weaker but systematic effects. The contribution of display type demonstrates substantially higher variance, indicating different sensitivity of display technologies to the artifacts.

## 7. Conclusion

We created a new diverse dataset containing videos compressed by various encoding standards, including HEVC, AV1, and VVC, and enriched with information about labeling conditions such as screen size, screen brightness, and ambient luminance. The labels were collected on more than 300 different devices. We used this dataset to analyze how both classical and modern learning-based objective quality metrics predict video ordering across different devices. Our analysis revealed that, due to human factor annotation noise and the strong dependence of pairwise preferences on viewing conditions, existing metrics achieve only limited accuracy in predicting quality orderings. To address this limitation, we proposed a training strategy for adapting and generalizing metrics to specific viewing conditions, which results in a clear improvement in ordering quality.

The proposed dataset will be valuable for researchers and practitioners developing image- and video-quality metrics for compression artifacts and optimizing solutions for diverse devices. It can be used to train models to perform VQA with higher accuracy and viewing condition specific precision, bringing more flexible encoders optimization.

## 8. Limitations

In this work, we did not retrain the evaluated metrics on the proposed dataset; instead, we applied the adaptation module to the outputs of pre-trained models without tuning their internal parameters. While this approach demonstrates the feasibility of condition-aware adaptation, it may limit the full potential of the underlying metrics. Future work will therefore include retraining or fine-tuning metrics directly on subsets of the dataset to achieve further improvements. Another limitation is that, in the current design, the adaptation module must be inferences separately for each target device, which may be inefficient when scaling to a large number of devices. As a next step, we plan to explore direct mappings from device distributions to score distributions, enabling more efficient and unified adaptation across diverse viewing conditions.

## Acknowledgment

The work of Nikolay Safonov was supported by the The Ministry of Economic Development of the Russian Federation in accordance with the subsidy agreement (agreement identifier 000000C313925P4H0002; grant No 139-15-2025-012).

The research was carried out using the MSU-270 supercomputer of Lomonosov Moscow State University. The labeling was performed using the Yandex Tasks platform.

## Impact Statement

This work introduces a multi-device video quality assessment (VQA) dataset and a condition-aware adaptation framework to model human visual perception across diverse real-world viewing environments. Environmentally, our approach enables streaming services to optimize adaptive bitrate algorithms based on specific display properties and ambient lighting, which can reduce unnecessary bandwidth consumption and the associated energy costs of large-scale video delivery. Ethically, our crowd-sourced data collection followed strict privacy protocols by excluding personally identifiable information (PII) and location data, while ensuring informed consent and fair compensation for all participants. Although the dataset is currently limited to the Android ecosystem and specific user demographics, which may introduce sampling biases, our proposed condition-pool augmentation strategy mitigates these limitations by mathematically simulating unobserved hardware states. Overall, this research poses no foreseeable risks of dual-use or malicious application, and instead provides a scientifically grounded methodology for developing more energy-efficient, equitable, and perceptually accurate video compression systems across heterogeneous consumer devices.

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

# A. Optimization Procedure

In this appendix, we provide a detailed derivation of the Expectation–Maximisation (EM) procedure used to jointly infer the latent subjective scores $\{q_i\}$ and optimise the parameters of the transformation networks $f_c(\cdot)$ and $f_b(\cdot)$.

## A.1. Complete-Data Likelihood

Let $\mathcal{D} = \{(i, j, z)\}$ denote the set of observed pairwise preference votes under viewing condition $z$, where $(i \succ j, z)$ indicates that video $i$ was preferred over video $j$. The probability of this observation under our model is

$$P(i \succ j, z \mid q, \theta, \psi) = \sigma\left(\|f_c(q_i, z; \theta) - f_b(q_j, z; \psi)\|^2 - \|f_c(q_j, z; \theta) - f_b(q_i, z; \psi)\|^2\right), \tag{4}$$

where $\sigma(\cdot)$ is the sigmoid function, and $\theta$ and $\psi$ are the parameters of $f_c$ and $f_b$, respectively.

The complete-data likelihood is then

$$\mathcal{L}_c(q, \theta, \psi \mid \mathcal{D}) = \prod_{(i,j,z)\in\mathcal{D}} P(i \succ j, z \mid q, \theta, \psi), \tag{5}$$

and the corresponding log-likelihood is

$$\log \mathcal{L}_c(q, \theta, \psi) = \sum_{(i,j,z)\in\mathcal{D}} \log \sigma\left(\Delta_{ij}(z; q, \theta, \psi)\right), \tag{6}$$

where

$$\Delta_{ij}(z; q, \theta, \psi) = \|f_c(q_i, z; \theta) - f_b(q_j, z; \psi)\|^2 - \|f_c(q_j, z; \theta) - f_b(q_i, z; \psi)\|^2. \tag{7}$$

## A.2. E-Step

In the E-step, we compute the expected log-likelihood with respect to the posterior of the latent variables $q$, given the current parameter estimates $(\theta^{(t)}, \psi^{(t)})$:

$$Q(q, \theta, \psi \mid \theta^{(t)}, \psi^{(t)}) = \mathbb{E}_{q\mid\mathcal{D},\theta^{(t)},\psi^{(t)}}\left[\log \mathcal{L}_c(q, \theta, \psi)\right]. \tag{8}$$

In practice, this expectation is approximated by point estimates of the latent scores $\{q_i\}$ obtained from the previous iteration, i.e. we set

$$q^{(t)} = \arg\max_q \log \mathcal{L}_c(q, \theta^{(t)}, \psi^{(t)}). \tag{9}$$

## A.3. M-Step

In the M-step, we maximise the surrogate function $Q$ with respect to both the latent scores and network parameters:

$$(q^{(t+1)}, \theta^{(t+1)}, \psi^{(t+1)}) = \arg\max_{q,\theta,\psi} Q(q, \theta, \psi \mid \theta^{(t)}, \psi^{(t)}). \tag{10}$$

This reduces to gradient-based optimisation of the log-likelihood. Specifically, the gradients are given by

$$\nabla_{q_i} \log \mathcal{L}_c = \sum_{(i,j,z)\in\mathcal{D}} (1 - \sigma(\Delta_{ij})) \nabla_{q_i}\Delta_{ij}(z; q, \theta, \psi), \tag{11}$$

$$\nabla_\theta \log \mathcal{L}_c = \sum_{(i,j,z)\in\mathcal{D}} (1 - \sigma(\Delta_{ij})) \nabla_\theta\Delta_{ij}(z; q, \theta, \psi), \tag{12}$$

$$\nabla_\psi \log \mathcal{L}_c = \sum_{(i,j,z)\in\mathcal{D}} (1 - \sigma(\Delta_{ij})) \nabla_\psi\Delta_{ij}(z; q, \theta, \psi). \tag{13}$$

The terms $\nabla_{q_i}\Delta_{ij}$, $\nabla_\theta\Delta_{ij}$, and $\nabla_\psi\Delta_{ij}$ can be computed via automatic differentiation since $f_c(\cdot)$ and $f_b(\cdot)$ are implemented as neural networks.

### A.4. Regularisation and Identifiability

Since the latent scores $\{q_i\}$ are only identifiable up to affine transformations, we impose constraints to avoid degeneracy:

$$\frac{1}{N} \sum_{i=1}^{N} q_i = 0, \qquad \frac{1}{N} \sum_{i=1}^{N} q_i^2 = 1. \tag{14}$$

These constraints normalise the latent quality scale, ensuring that the scores are comparable across training runs.

### A.5. Summary

The EM optimisation alternates between:

1. Updating the latent quality scores $\{q_i\}$ based on the current network parameters (E-step),

2. Updating the network parameters $(\theta, \psi)$ by maximising the surrogate log-likelihood (M-step),

until convergence. In practice, we implement both steps jointly using stochastic gradient descent, with normalisation of $\{q_i\}$ applied after each iteration to enforce identifiability.

## B. Dataset

### B.1. Videos Preparation

We have 18,000 open-source high-bitrate videos licensed under CC BY or CC0. The clips include both professional and amateur recordings, with nearly half exhibiting scene changes and high motion. The proportion of synthetic to natural lighting is approximately 1:3. The content covers a wide range of categories, including nature, sports, close-up human scenes, gameplay, music videos, water or steam, and CGI. Common effects and distortions include camera shake, slow motion, grain and noise, extreme lighting conditions, on-screen text, and extraneous objects near the camera lens. This diversity enables a more realistic simulation of real-world viewing conditions.

For the SI/TI calculation and building a representative set samples were encoded using x264 with a constant quantization parameter, after which the spatial and temporal complexity of each scene was computed. Spatial complexity is defined as the average I-frame size normalized by the uncompressed frame size, while temporal complexity is computed as the ratio of the average P-frame size to the average I-frame size. In addition, a preprocessing step was applied to standardize chroma subsampling across all videos, as it affects complexity estimation. Specifically, all sequences were converted to YUV 4:2:0.

### B.2. Crowd-Worker Protocol

This study involves human participants who performed subjective video quality assessment (VQA) tasks using their personal Android devices. The data collection was conducted with careful consideration of ethical principles, including informed consent, fair compensation, data minimization, and participant privacy.

Participants were recruited via a crowd-sourcing platform and voluntarily installed a custom Android application developed specifically for this study. Before participation, users were presented with a clear description of the task, the types of data being collected, and the purpose of the study. Participation was fully optional, and users could withdraw at any time by uninstalling the application, after which no further data were collected.

The application performed a subjective VQA task in which participants compared or rated video content under natural viewing conditions. In addition to subjective responses, the application logged limited device and environmental metadata required for the analysis of viewing conditions, including device model and basic hardware specifications, as well as screen brightness and ambient light levels sampled once per second during the task. No audio, video, location, contact information, or other sensitive personal data were collected.

All collected data were pseudonymized at the source using randomly generated participant identifiers. No directly identifying information (such as names, phone numbers, email addresses, advertising IDs, or precise location data) was stored. Device metadata were limited strictly to attributes necessary for grouping results by display characteristics and viewing conditions.

*Table 4.* Encoding commands used for creating compressed videos

| Encoder (name, version) | Encoding commands |
| --- | --- |
| x265 (2020-04-13) | `x265.exe tune ssim preset medium crf TARGET_RC SOURCE_FILE -o TARGET_FILE input-res WIDTHxHEIGHT fps FPS` |
| rav1e (unspecified) | `rav1e.exe threads 12 tiles 8 min-keyint 25 keyint 250 speed 3 quantizer TARGET_RC -o TARGET_FILE.ivf SOURCE_FILE` |
| SVT-HEVC (v1.5.1) | `SvtHevcEncApp.exe -i SOURCE_FILE -w WIDTH -h HEIGHT -fps FPS -rc 1 -tbr TARGET_BITRATE -encMode 5 -b TARGET_FILE` |
| SVT-AV1 (v0.8.3) | `SvtAv1EncApp.exe -i SOURCE_FILE -w WIDTH -h HEIGHT fps FPS rc 0 -q BITRATE_KBPS preset 3 -b TARGET_FILE` |
| SVT-VP9 (v0.2.0) | `SvtVp9EncApp.exe -i SOURCE_FILE w WIDTH h HEIGHT fps FPS rc 0 -q BITRATE_KBPS -enc-mode 0 -b TARGET_FILE` |
| x265 (3.5+1-ce882936d) | `x265-8bit.exe input SOURCE_FILE input-res WIDTHxHEIGHT fps FPS crf TARGET_RC vbv-bufsize 12000 vbv-maxrate 12000 preset veryslow rskip 0 ref 5 merange 92 rc-lookahead 50 -o TARGET_FILE psnr ssim tune=ssim` |
| vvenc (v1.0.0) | `./vvencapp preset fast -i SOURCE_FILE -s WIDTHxHEIGHT -r FPS bitrate TARGET_BITRATE -o TARGET_FILE` |
| rav1e (0.5.0-alpha (p20210518)) | `rav1e-ch.exe threads 16 tiles 8 slots 2 min-keyint 25 keyint 250 speed 3 quantizer TARGET_RC frame-rate FPS_NUM time-scale FPS_DENOM -o TARGET_FILE.ivf SOURCE_FILE` |

Raw sensor logs were retained only for the duration required for statistical analysis and were stored on secure servers with restricted access.

Participants were compensated monetarily for their time. Task design and payment rates were calibrated to ensure that the effective hourly compensation met or exceeded local minimum wage estimates for crowd work, accounting for task duration, installation time, and potential overhead. No unpaid screening or mandatory follow-up tasks were required.

While the study did not involve medical procedures or interventions, it constitutes human-subjects research due to the collection of subjective responses and contextual metadata. An ethics review focused on informed consent, proportional data collection, and privacy safeguards is therefore warranted. The study design follows widely accepted ethical guidelines for crowd-sourced user studies and low-risk human-computer interaction research, emphasizing transparency, voluntariness, and protection of participant data.

### B.3. Data Processing

To evaluate the proposed methods under realistic viewing conditions, we use a dedicated mobile application and a crowd-sourcing platform to recruit participants. Only users who explicitly agree to run the application and follow the testing instructions are selected. When a user launches the application, the device model is automatically recorded. The application starts a video quality assessment task defined by our experimental protocol. During the entire session, the application continuously monitors several important parameters that may influence visual perception. In particular, if the device provides an ambient light sensor, the current illumination level is logged. This information is later used to analyze or filter sessions conducted under unsuitable lighting conditions. Ambient light sensor readings collected during the experiments were additionally filtered to remove unreliable measurements. First, we excluded sessions from devices that did not provide a dedicated ambient light sensor or reported constant illumination values over time, which typically indicates missing or non-functional sensors. Second, we removed measurements with abrupt and physically implausible changes in illumination, such as large spikes or drops occurring within one-second intervals, as these are often caused by sensor noise, temporary occlusions, or background system events. Finally, sessions with illumination values consistently outside a predefined valid range were discarded. This filtering procedure ensures that only stable and meaningful ambient light measurements are used in the analysis, reducing noise and preventing sensor artifacts from influencing the reported results.

To ensure consistent viewing conditions, the application enforces a horizontal (landscape) screen orientation throughout the test. If the orientation changes, the session is paused or discarded. In addition, the application tracks the screen brightness once per second. Since different devices use different brightness scales (commonly ranging from 0–100 or 0–255), the recorded brightness values are normalized to the range [0,1] using the device-specific maximum brightness value obtained from system settings. We incorporated a server-controlled option in the application that allows the screen brightness to be set to predefined levels. Using this mode, we collected additional test sessions at the minimum and maximum brightness levels supported by each user device. This procedure enables controlled analysis of visual quality under extreme brightness conditions while accounting for device-specific brightness limits.

All collected signals are logged together with the subjective responses. To the final dataset mean and maximum results are reported. Based on the device model we have filled the dataset with device properties obtained from the web. Figures 6 and 7 demonstrate the distribution of the models in the testing. Table 5 lists the available metadata fields.

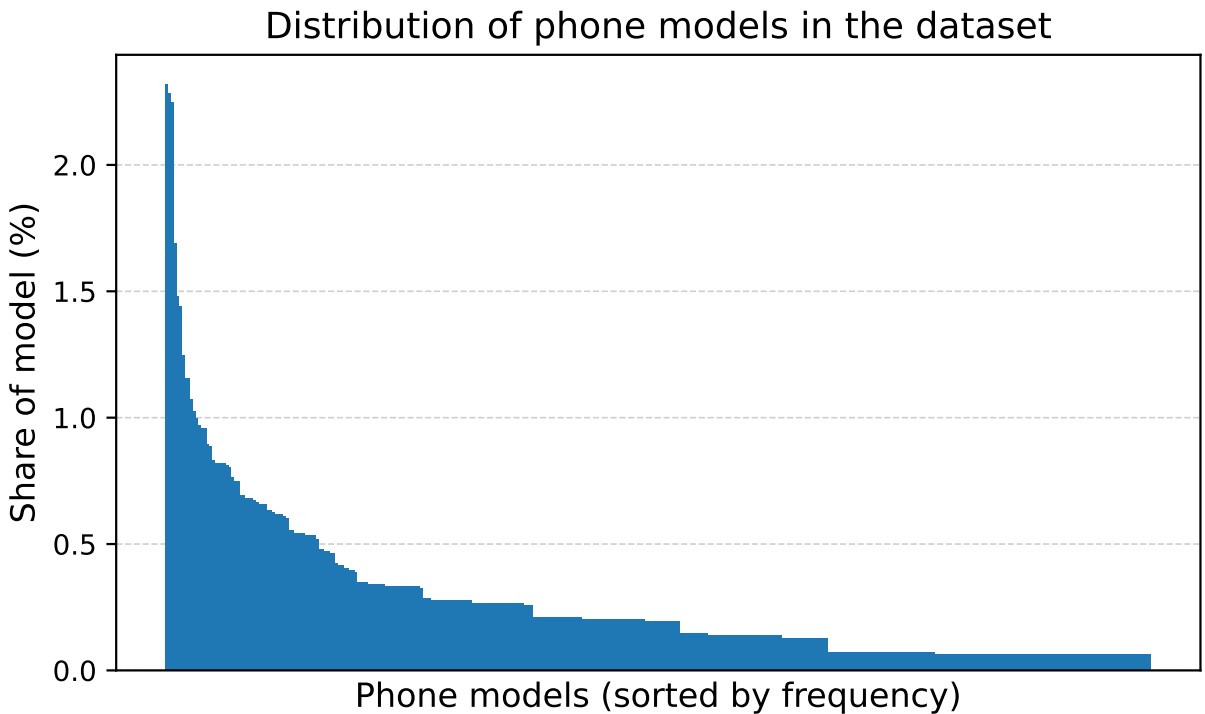

*Figure 6.* Distribution of the phone models frequency.

## C. Evaluation

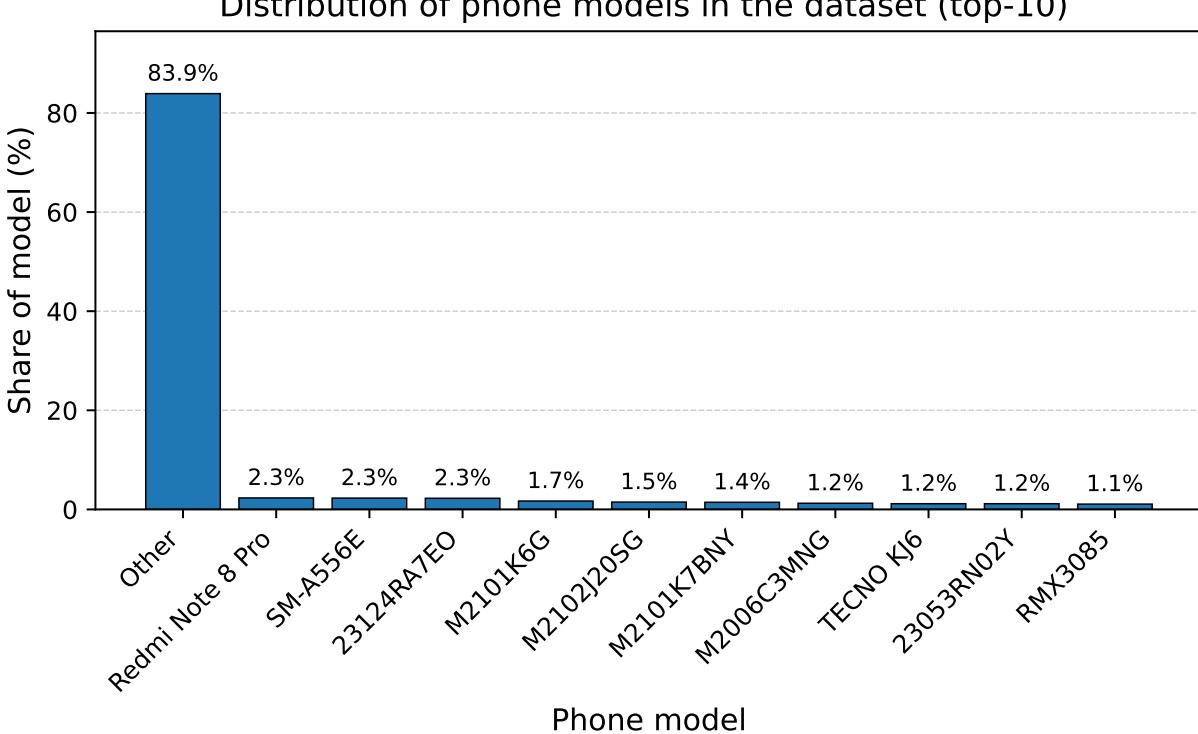

*Figure 7.* Distribution of the phone models frequency. Top-10 models cover almost 20% of the votes.

*Table 5.* Description of the display and environment characteristics collected during testing.

| Name | Data type | Description |
|---|---|---|
| h_pix | integer | Display height in pixels |
| w_pix | integer | Display width in pixels |
| dpi | integer | Display pixel density (dots per inch) |
| h_ph | float | Physical display height in inches |
| w_ph | float | Physical display width in inches |
| model | string | Device model name |
| displaysize | float | Display diagonal size in inches |
| HDR | boolean | HDR support flag |
| Dolby Vision | boolean | Dolby Vision support flag |
| Display type | categorical | Display technology (e.g., OLED, LCD) |
| Peak (nits) | float | Peak display luminance in nits (sparse) |
| HBM (nits) | float | High Brightness Mode luminance in nits (sparse) |
| Typical (nits) | float | Typical display luminance in nits (sparse) |
| Hz | integer | Display refresh rate in Hz |
| bit-depth | integer | Display color bit depth |
| illumination_max | float | Maximum ambient light level during session |
| illumination_mean | float | Mean ambient light level during session |
| has_light_sensor | boolean | Indicates presence of ambient light sensor |
| brightness_max | float | Maximum recorded screen brightness (normalized) |
| brightness_mean | float | Mean screen brightness (normalized) |

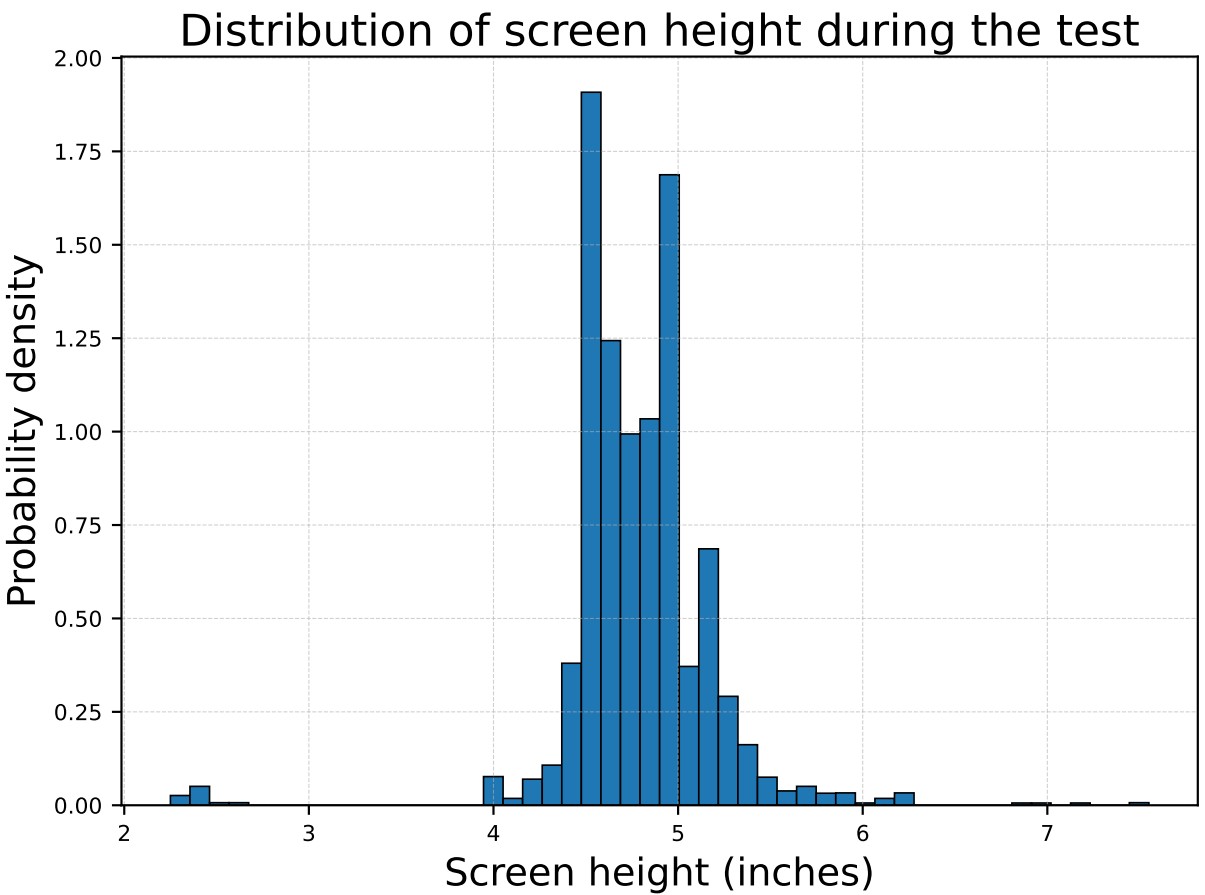

*Figure 8.* Distribution of device heights.

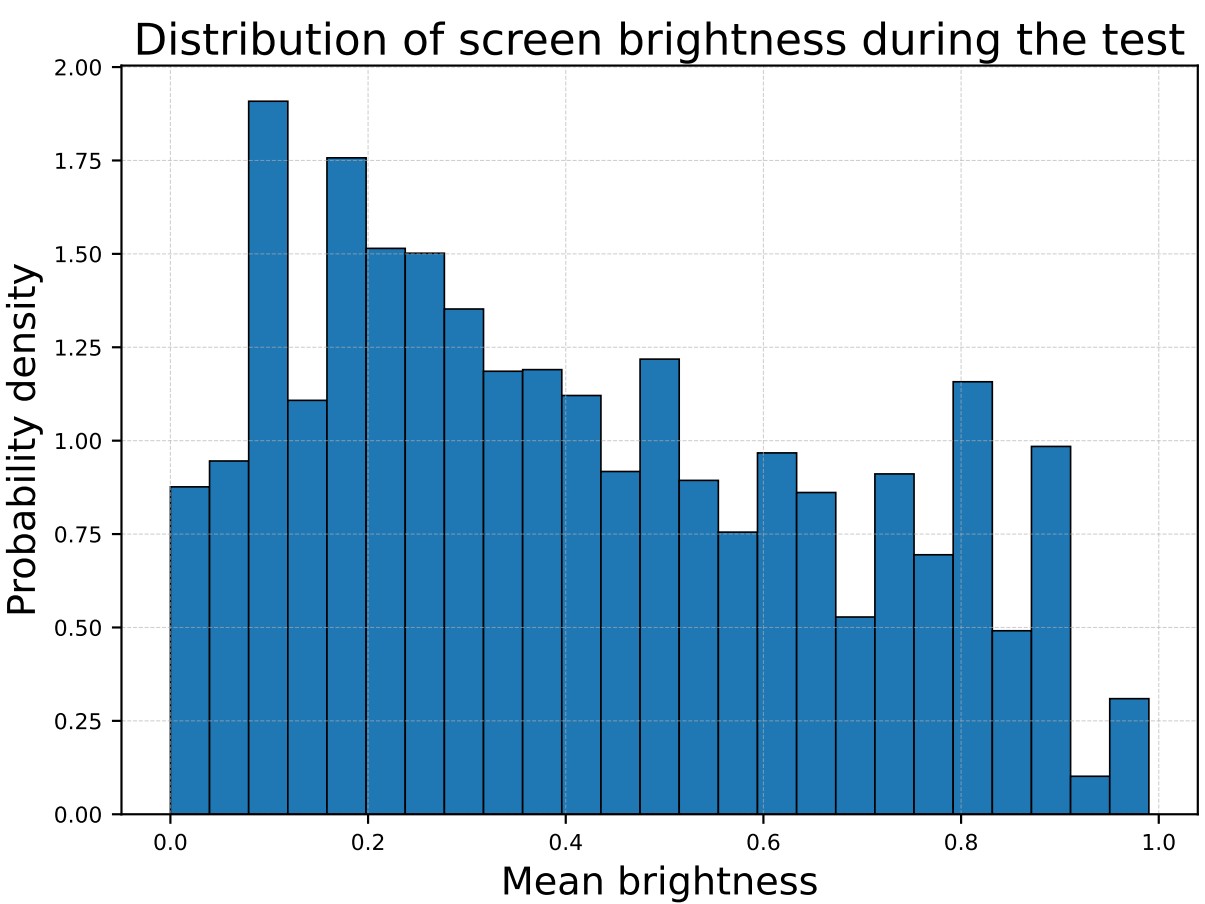

*Figure 9.* Distribution of screen brightness levels.

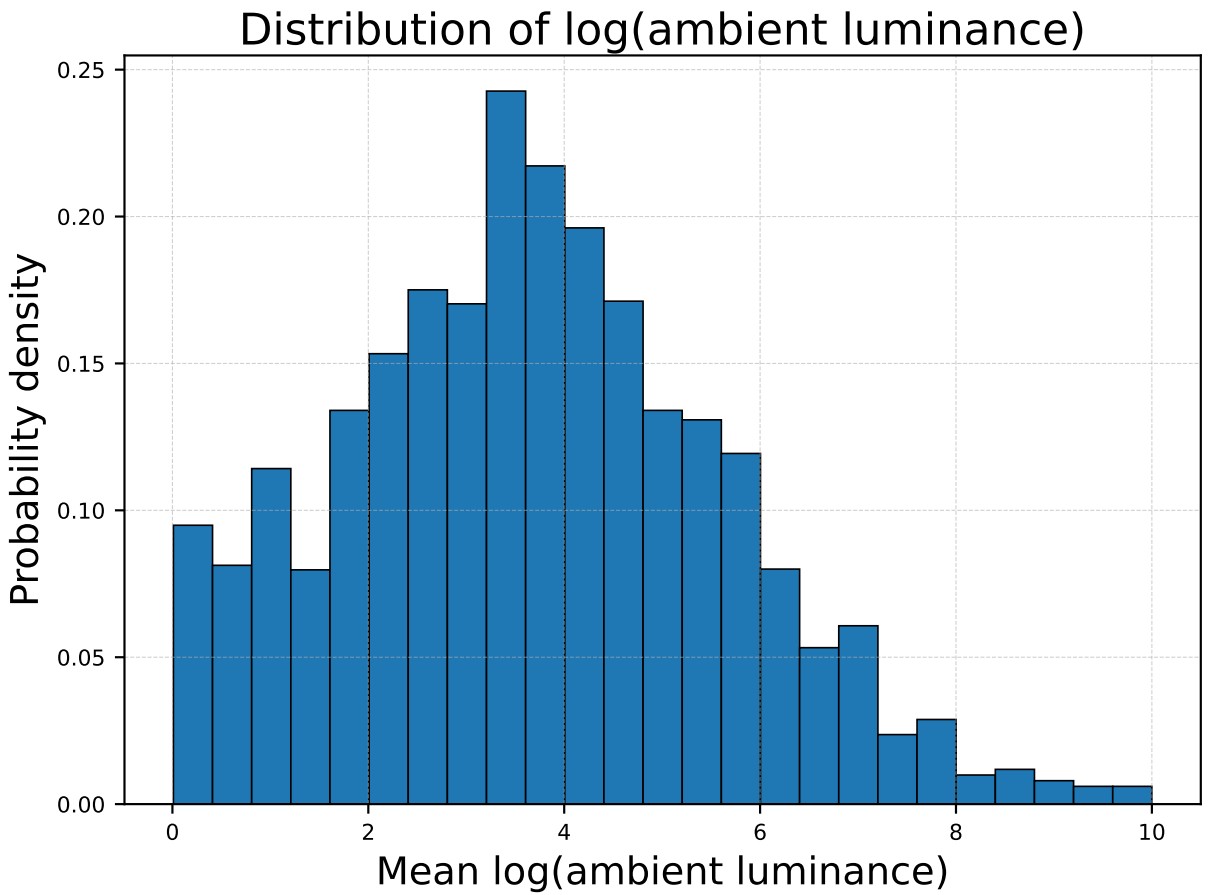

*Figure 10.* Distribution of the ambient lightning.

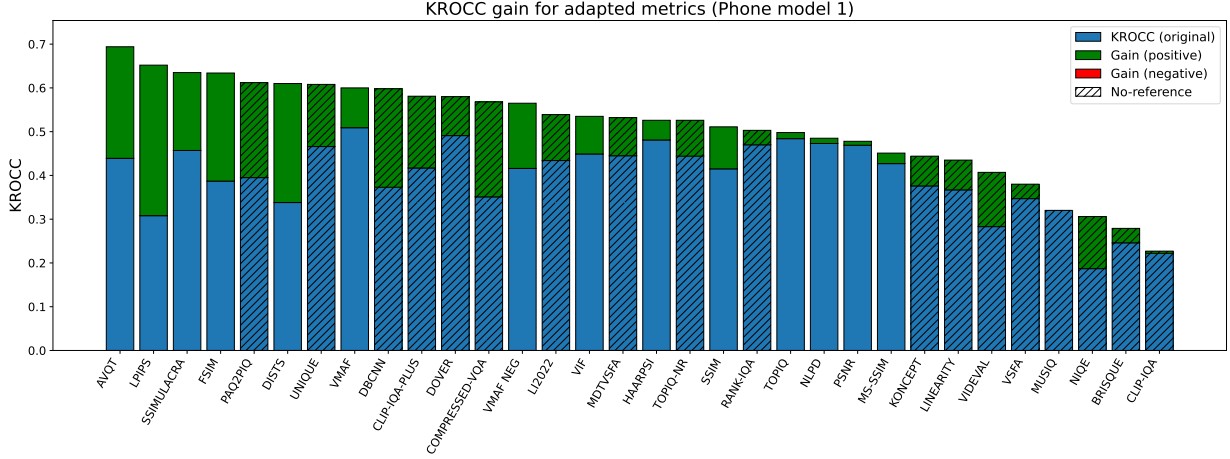

*Figure 11.* Kendall rank correlations for the original metrics and their adapted counterparts for Xiaomi Redmi Note 13. Gain represents how score improved after metric training (green = positive, red = negative)

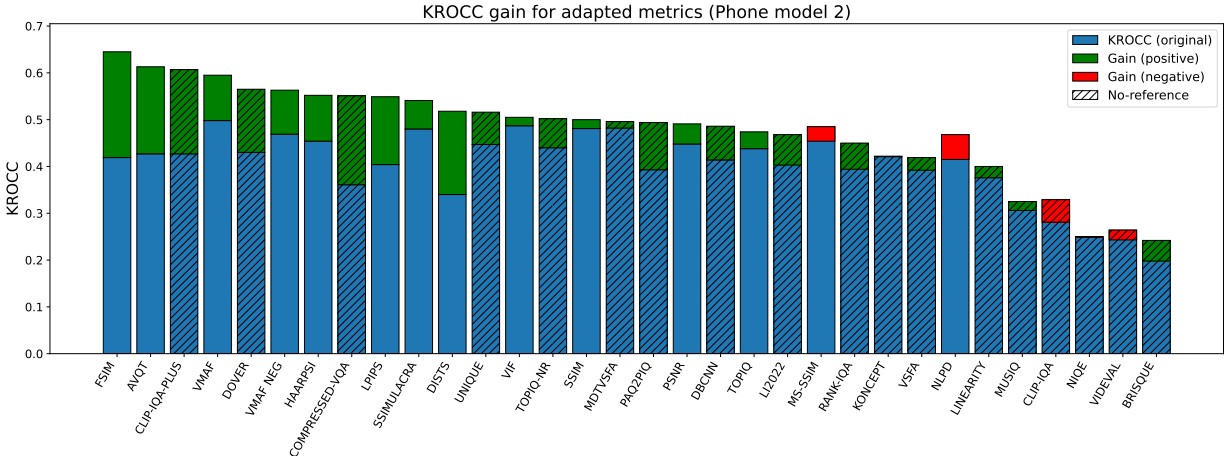

*Figure 12.* Kendall rank correlations for the original metrics and their adapted counterparts for Samsung Galaxy A55. Gain represents how score improved after metric training (green = positive, red = negative)

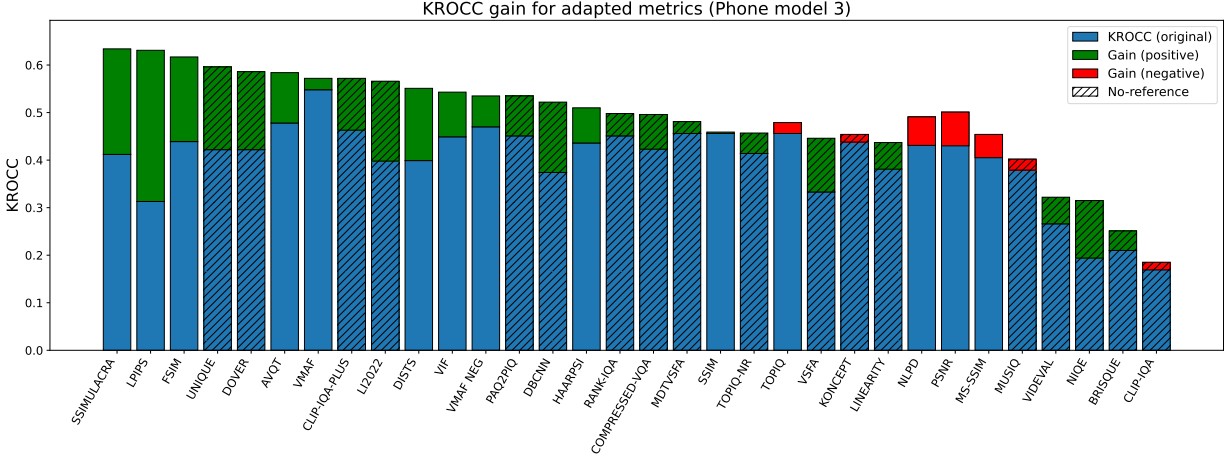

*Figure 13.* Kendall rank correlations for the original metrics and their adapted counterparts for Xiaomi Redmi Note 8 Pro. Gain represents how score improved after metric training (green = positive, red = negative)

