# OpenReview forum: "Learning Flexible Generalization in Video Quality Assessment by Bringing Device and Viewing Condition Distributions"
_ICML.cc/2026/Conference — ICML 2026 regular_

### Official Review · Reviewer_cwxt · 2026-03-09

**Soundness:** 3
**Presentation:** 3
**Significance:** 3
**Originality:** 3
**Overall Recommendation:** 4
**Confidence:** 4

**Summary:**

This paper studies condition-aware video quality assessment where perceived quality depends not only on compression artifacts but also on device and viewing conditions such as screen size, brightness, HDR capability, and ambient light.
The authors collect a large crowd-sourced multi-device dataset and propose a condition-aware aggregation model based on Blade–Chest embeddings to infer latent subjective quality scores.

**Compliance With Llm Reviewing Policy:**

Affirmed.

**Final Justification:**

The rebuttal strengthens the work with additional clarifications and experiments, though some concerns about real-world generalization and competitiveness remain. Overall, the paper is promising and I will maintain my score of weak accept.

**Key Questions For Authors:**

1. Given the flexible nonlinear transforms \(f_c\) and \(f_b\), what guarantees exist that the latent scalar \(q_i\) represents an intrinsic device-invariant quality score rather than simply an index used by the embedding networks?

2. How does the proposed adaptation module compare to simpler baselines such as affine calibration, generalized additive models, or device-specific regression layers applied on top of the base VQA metric?

**Limitations:**

The approach assumes access to device metadata and viewing conditions at inference time, which may not always be reliably available in real deployment scenarios.

**Strengths And Weaknesses:**

## Strengths
1. The paper highlights a fundamental limitation of standard VQA pipelines: perceptual quality ordering is not invariant across viewing devices. Modeling quality as a function of both content distortion and device/viewing context is technically meaningful and relevant for real-world deployment.

2. The dataset contains ~250K pairwise comparisons collected from ~10K users across more than 300 Android devices with recorded metadata (brightness, ambient light, screen characteristics). This provides a rare large-scale benchmark for studying cross-device perceptual variability.

## Weaknesses
1. The EM optimization procedure and identifiability of the latent decomposition are not clearly justified.

2. The paper does not compare against simpler condition-aware baselines such as linear calibration, device-specific bias terms, or Bradley–Terry models with metadata features. As a result, it is unclear whether the complex Blade–Chest formulation is necessary.

---

> ### Author Rebuttal · Authors · 2026-03-31
>
> We sincerely appreciate the reviewer’s thoughtful feedback and constructive recommendations. These will be incorporated into the revised manuscript. We provide detailed responses to the raised concerns below.
>
> **W1**. The EM optimization is used as a practical approach to jointly estimate the latent quality scores and the parameters of the condition-dependent transformations under pairwise supervision. While the latent decomposition itself is not strictly identifiable in an absolute sense (as is common in latent variable models such as Bradley–Terry), our goal is not to recover uniquely interpretable embeddings, but to obtain a consistent ordering of samples under varying conditions. We will revise the paper to make this distinction explicit: the goal of the EM procedure is to obtain a stable conditional decomposition that maximizes the pairwise likelihood. Use of EM is motivated by the fact that the latent quality scores $q_i$ are not directly observed, while supervision is available only through pairwise votes under conditions $z$. The supplementary makes this explicit by defining the complete-data likelihood, the E-step as updating the latent scores under the current network parameters (current conditions), and the M-step as updating the parameters of $f_c$ and $f_b$ through maximization of the surrogate objective.
>
> **W2**. **These approaches are limited by the sparsity of labeled data across diverse viewing conditions**, which makes reliable learning of condition-dependent effects challenging. Moreover, for certain classes of models, there is a theoretical limitation in their formulation that prevents them from explicitly incorporating or learning the influence of viewing conditions $z$.
> The condition pool is motivated by the sparsity of subjective data across heterogeneous viewing conditions. After obtaining latent scores and condition-aware transformations via Blade–Chest aggregation, we decouple condition sampling from the original annotations, enabling effective data augmentation and improving generalization, including to unseen conditions.
>
> Importantly, this mechanism relies on the Blade–Chest formulation, which explicitly models viewing conditions during aggregation. In contrast, standard approaches (e.g., Bradley–Terry, pairwise logistic regression, GAM) define the matchup function based only on item differences and assume condition-invariant preferences, making them unsuitable for condition-aware resampling. Let $M(x_i, x_j)$ denote a matchup function. This function can be defined in a general form, provided that the following conditions hold:
> * $M(x_i, x_j) \in \mathbb{R}$; positive values indicate that item $i$ is more likely to win, while negative values indicate the opposite. When $M(x_i, x_j) = 0$, both outcomes are equally likely;
> * as $M(x_i, x_j) \to +\infty$, $P(x_i \succ x_j) \to 1$, and vice versa;
> * $M(x_i, x_j) = -M(x_j, x_i)$, ensuring consistency of pairwise probabilities.
>
> Within this framework, classical models can be expressed as specific choices of $M$. For example, generalized additive models (GAM) take the form
> $$
> M(x_i, x_j) = \sum_k g_k\big(\phi_k(x_i) - \phi_k(x_j)\big),
> $$
> where $g_k(\cdot)$ are learned univariate functions. As a result, when two videos are compared under the same viewing conditions, the condition-dependent factors cancel out for the model.
>
> While it is possible to attempt to learn such dependencies directly with standard models, we found this to be ineffective in practice due to the sparsity of condition coverage. We will include these results in the supplementary material.
>
> **Q1**. We apologize for the confusion, but we are not fully certain that we correctly understand the question. Could you please clarify what is meant by the “index used by the embedding networks”?. The role of $f_c$ and $f_b$ is to model condition-dependent deviations from this shared scale. We validate that $q_i$ captures a meaningful, device-invariant notion of quality by showing high correlation with Bradley-Terry scores and reference (desktop) scores. If $q_i$ were merely an arbitrary index used by the embedding networks, such consistency across aggregation methods and conditions would not be observed.
>
> **Q2**. Please, refer to W1.
>
> If you have no further concerns we would be sincerely grateful if you could consider raising the rating of our submission.

---

> > ### Author Rebuttal · Reviewer_cwxt · 2026-04-02
> >
> > Thanks for adding the additional experiments and stronger baselines; these address my main concerns. I will maintain my original score of 4.

---

> > > ### Author Response · Authors · 2026-04-08
> > >
> > > We thank the reviewer for their careful evaluation and greatly appreciate the constructive discussion. We believe that this work provides meaningful contributions to the field by introducing a novel multi-screen formulation, a large-scale dataset, and an effective learning paradigm. Therefore, as all the reviewer’s concerns have been addressed, we would kindly ask them to reconsider their score.

---

### Official Review · Reviewer_piPr · 2026-03-11

**Soundness:** 2
**Presentation:** 3
**Significance:** 3
**Originality:** 2
**Overall Recommendation:** 3
**Confidence:** 4

**Summary:**

This paper tackles multi-screen video quality assessment by creating a large-scale crowdsourced dataset with 250K+ pairwise comparisons across 300+ Android devices, enriched with metadata (screen specs, brightness, ambient light). The authors propose a Blade-Chest-based adaptation module that incorporates viewing conditions into quality score aggregation and a training strategy for generalization. Evaluation shows existing VQA metrics perform poorly across conditions, while the adaptation substantially improves correlations (e.g., LPIPS from 0.308 to 0.652 KROCC). This is the first framework explicitly training metrics to account for device characteristics.

**Compliance With Llm Reviewing Policy:**

Affirmed.

**Key Questions For Authors:**

Please refer to the strengths and weaknesses.

**Limitations:**

Please refer to the strengths and weaknesses.

**Strengths And Weaknesses:**

**Strengths:**
- Impressive dataset scale with rich metadata addressing a genuine research gap
- Comprehensive evaluation of 30+ existing metrics
- Clear demonstration that viewing conditions significantly impact perception

### Weaknesses
- **Statistical reliability of subjective scores**: While the paper collects 15+ judgments per pair *globally*, it is unclear how many independent assessors contribute to each *device-condition* subset. ITU-R BT.500 and ITU-T P.910 recommend 15–24+ subjects *per testing condition* for reliable MOS estimation. The crowd-sourced design may introduce high variance that is not fully addressed.
- **Oversimplified adaptation module**: The adaptation network is a multi-layer MLP that takes metric output + condition vector as input. This design treats viewing conditions as auxiliary features rather than modeling how conditions *interact* with video content characteristics (e.g., how banding artifacts become more visible on high-brightness HDR screens, or how motion complexity affects perception on small screens). A more content-condition interaction mechanism could yield stronger gains.
- **Insufficient analysis of condition-content interactions**: The paper shows correlation improvements but lacks deeper analysis of *which* artifacts or content types are most sensitive to specific conditions. For example, how does resolution interact with screen size? Does ambient light affect dark-scene quality more than bright scenes? Such insights would strengthen the practical value of the work.
- **Dataset quality control under heterogeneous conditions**: While the app logs metadata, the paper does not detail how it ensures that variations in viewing conditions are *meaningful* and not confounded by uncontrolled factors (e.g., viewing distance, user attention, sensor calibration drift). More rigorous filtering or stratification protocols would improve confidence in the labels.

---

> ### Author Rebuttal · Authors · 2026-03-31
>
> We are grateful for the reviewer’s detailed feedback and helpful suggestions. We will take them into account in the revised version of the paper. Below, we address the raised concerns:
>
> **W1**. It should be noted that our study **does not rely on MOS collection, but on pairwise preference judgments**. Unlike MOS, which requires subjects to assign an absolute quality score and is therefore prone to individual scaling bias, pairwise comparisons impose a significantly simpler and more reliable task, as selecting the better video or indicating no visible difference is easier. This formulation reduces cognitive load and inter-subject bias, making it more suitable for large-scale crowd-sourced settings. Each pairwise comparison is treated as an independent observation, and condition effects are explicitly modeled rather than averaged out.
>
> At the same time, we acknowledge that subjective labels remain inherently noisy, especially under diverse uncontrolled conditions. This is reflected in the moderate KROCC values observed even after adaptation. This was noted in the paper Evaluation section.
>
> **W2**. As it has been pointed out in the previous section our formulation is fundamentally based on pairwise preference judgments rather than MOS scores. This has important implications for the type of analysis that can be reliably performed. For example, one video may contain blocking artifacts (e.g., from H.264), while another exhibits blur (e.g., from AV1), and the final preference depends on the relative strength and perceptual impact of these distortions under given conditions. As a result, the **observed label does not correspond to a specific artifact type**, making direct modeling or attribution of condition–artifact interactions inherently ambiguous.
>
> Given this setting, the role of the adaptation module is not to disentangle individual artifact sensitivities, but to learn a robust conditional mapping from metric predictions to perceived quality under varying viewing conditions. The MLP design provides a flexible and stable way to capture nonlinear dependencies between predicted quality and observable conditions
>
> **W3**. As we point out in the next section, rather than analyzing individual preferences, it is more appropriate in our setting to consider aggregated scores and improvements in metric prediction. Therefore, we conducted the following experiment: we selected 5 darkest and 5 brightest videos based on average luma level, and evaluated adaptation for VMAF and DOVER. The results show higher gains for the darkest videos, with improvements of 10% and 7%, respectively. We will add a more detailed analysis of this case to the supplementary materials.
>
> **W4**. We do not treat viewing conditions as an uncontrolled nuisance variable, but as an explicitly observed covariate $z$ that is accepted only if it is measurable, temporally stable, and physically plausible, and then validated through its explanatory power.
> We agree that certain factors (e.g., viewing distance, user attention, and sensor calibration drift) are not explicitly controlled and may influence subjective preferences. However, **these factors are inherently uncontrolled in real-world industrial applications, where such information is typically unavailable to service providers**. Therefore, our goal is to address a practically relevant setting by modeling observable and measurable aspects of the viewing environment, demonstrating that incorporating the distribution of user devices and conditions enables more flexible and realistic quality estimation. In this sense, **our approach can be viewed as a natural extension of traditional VQA frameworks**, which largely ignore even these observable factors, toward a more deployment-oriented and condition-aware formulation.
>
> From a modeling perspective, the meaningfulness of the retained variability is validated by testing whether conditioning on $z$ affects the preference structure. In our formulation, pairwise preferences are modeled as
> $$
> P(i \succ j \mid z) = \sigma\ \left(
> \|f_c(q_i, z) - f_b(q_j, z)\|^2 -
> \|f_c(q_j, z) - f_b(q_i, z)\|^2
> \right),
> $$
> where $z$ explicitly modulates the comparison. If the recorded conditions were dominated by noise or confounding factors, incorporating $z$ would not improve predictive consistency. However, **we observe systematic and physically interpretable effects (e.g., monotonic shifts across screen size strata, differences between display technologies discrepancies), as well as consistent improvements in rank correlation after conditioning and adaptation**. This demonstrates that the variation in $z$ captures reproducible perceptual effects rather than uncontrolled artifacts. We note that pointwise statistical tests are not reliable in this setting due to the inherent noisiness of individual labels, as discussed earlier.
>
> If your concerns are resolved, we would greatly appreciate your consideration in updating the rating of our submission.

---

> > ### Author Rebuttal · Reviewer_piPr · 2026-04-03
> >
> > While the clarification regarding pairwise preference formulation is helpful, it does not fully alleviate my concern about the lack of explicit content–condition interaction modeling in the adaptation module; therefore, I retain my original score

---

> > > ### Author Response · Authors · 2026-04-08
> > >
> > > We thank the reviewer for the thoughtful follow-up and valuable suggestion. To further investigate the potential content dependency of artifacts, we conducted additional experiments incorporating commonly used video complexity and diversity descriptors, including spatial information (SI), temporal information (TI), colorfulness, average luminance, and contrast.
> > >
> > > These features were added to the predictor both individually and in combination (SI+TI and all features jointly). The results for several representative metrics are summarized in Table X. As can be seen, incorporating these features consistently improves model performance, confirming the relevance of content-dependent factors. We appreciate this suggestion, as it led to a meaningful extension of our analysis.
> > >
> > >
> > > Metric        | Gain   | +SI    | +TI    | +Color | +Luma  | +Contr | +SI+TI | +ALL
> > > --------------|--------|--------|--------|--------|--------|--------|--------|--------
> > > LPIPS         | 0.229  | 0.242  | 0.238  | 0.206  | 0.233  | 0.240  | 0.253  | 0.256
> > > DISTS         | 0.171  | 0.177  | 0.181  | 0.175  | 0.173  | 0.170  | 0.186  | 0.184
> > > VMAF          | 0.163  | 0.168  | 0.169  | 0.155  | 0.163  | 0.170  | 0.173  | 0.175
> > > SSIM          | -0.001 | 0.002  | 0.001  | 0.000  | -0.001 | 0.001  | 0.001  | 0.002
> > > DOVER         | 0.109  | 0.118  | 0.124  | 0.110  | 0.113  | 0.120  | 0.122  | 0.126
> > >
> > >
> > > Following these findings, we have incorporated the extended formulation into the revised version of the paper. Overall, we believe that the proposed dataset and learning framework provide substantial value for both research and practical applications in video quality assessment. We thank the reviewer again for helping us strengthen the work, and kindly ask them to reconsider their score if their concerns have been addressed.

---

### Official Review · Reviewer_Hr6s · 2026-03-12

**Soundness:** 3
**Presentation:** 2
**Significance:** 3
**Originality:** 4
**Overall Recommendation:** 4
**Confidence:** 4

**Summary:**

A large-scale dataset of multi-screen video quality is introduced, spanning 326 unique Android devices under various viewing conditions to address a gap in existing research. The dataset comprises over 250K pairwise preference judgments along with rich metadata including device model, screen technology, diagonal size, peak brightness, applied brightness settings, and ambient light measurements. Analysis shows that current objective quality metrics provide poor prediction accuracy compared to subjective opinions, as pairwise preferences are strongly contingent on different platforms and viewing conditions. The paper analyzes limitations of modeled image quality in widely-used metrics and introduces a training strategy using the Blade-Chest model that adapts existing metrics based on target viewing conditions by learning transformation functions conditioned on quality scores and environmental parameters. The adapted metrics yield improvements in Kendall rank correlations of up to 0.344 for LPIPS across several phone models and consistently show improvements for most evaluated metrics. The findings demonstrate that device and context information, when included in the model, greatly improves both accuracy and robustness of predictions across viewing conditions.

**Compliance With Llm Reviewing Policy:**

Affirmed.

**Key Questions For Authors:**

The adaptation module is applied to metric outputs rather than retraining the metrics themselves. Would fine-tuning the underlying metrics on the dataset yield further improvements, and are there plans to release fine-tuned versions?

How computationally intensive is this approach when scaling to thousands of devices? Is there a path toward learning a unified mapping from device characteristics to quality adjustments rather than device-specific adaptations?

The dataset includes both SDR and HDR content viewed on different displays. Were there systematic differences in how metrics performed on HDR versus SDR content, and does the adaptation handle both equally well?

The analysis shows that some metrics (e.g., PSNR, MS-SSIM) occasionally worsen after adaptation. What characteristics distinguish metrics that benefit from adaptation from those that do not?

Participants were compensated and the data involved privacy considerations. Could the authors elaborate on the ethical review process and measures taken to protect participant privacy beyond the brief description in Appendix C.2?

**Limitations:**

Yes, the authors discuss limitations in Appendix A. They acknowledge that metrics were not retrained on the dataset, only adapted at the output level, which may limit potential improvements. They also note that the adaptation module must be inferred independently for each target device, which can be inefficient at scale. Future work directions are suggested, including retraining metrics on subsets and exploring direct mappings from device distributions to score distributions. Ethical considerations of the crowdsourcing study are addressed in Appendix C.2, covering informed consent, data minimization, pseudonymization, and fair compensation. A dedicated limitations section in the main paper would improve presentation.

**Strengths And Weaknesses:**

Soundness: The methodology is solid and well-executed. Dataset construction is rigorous, including clustering in spatial-temporal complexity space, multiple compression standards (HEVC, VVC, AV1), and effective crowdsourcing protocols with control pairs and data filtering. The Blade-Chest model adaptation is well-defined with clear mathematical formulation and EM optimization. Extensive testing on 30+ metrics across multiple phones with clear reporting of gains is presented. Ablation and SHAP analysis provide additional validation. However, the adaptation module is applied to metric outputs rather than retraining the metrics themselves, which may limit potential gains. The computational cost of running different adaptations per device is not discussed.

Presentation:  The paper is well-structured and easy to read. The motivation is clearly illustrated with examples of quality perception differences across devices. Tables are comprehensive and effectively summarize the extensive experimental results. The data clearly shows distributions of device types, brightness levels, and ambient light. The mathematical formulation is clear.

Relevance:  The problem is extremely relevant to both academia and industry. Video quality assessment plays a core role in streaming services, content delivery, and device optimization. The multi-screen angle connects directly with the reality of varied consumer devices that current datasets mostly overlook. The results have real-world implications for adaptive streaming, encoder optimization, and quality monitoring. The dataset itself is an invaluable resource that will enable future research. The work advances the field by demonstrating that device context is more than noise and can be used for informing quality predictions.

Originality: The work is genuinely original. While video quality datasets exist, this is the first large-scale multi-device dataset with rich in-the-wild viewing condition metadata as opposed to controlled lab-based measurements. The key contribution is the application of the Blade-Chest model for condition-aware metric adaptation. The systematic analysis of how different display technologies (LED vs LCD, HDR vs SDR) impact quality perception yields new insights. The dataset scale (300+ devices, 250K annotations) is unprecedented in this domain. The Turnitin check reveals no concerning overlap with existing work.

---

> ### Author Rebuttal · Authors · 2026-03-31
>
> Thank you for your valuable suggestions and thoughtful feedback. We will use suggestions to enhance the revised version of the paper. Your questions are answered below:
>
> **Q1**. We conducted additional experiments after submission to evaluate fine-tuning of the underlying metrics on the proposed dataset. While fine-tuning does lead to further improvements, the gains are relatively modest and do not significantly exceed those achieved by the adaptation module alone. For example for VMAF it gives additional 0.019 gain, and for DOVER 0.014 gain.
> We hypothesize that this is primarily due to the inherent noise in the subjective labels, which limits the achievable performance and leads to an early saturation point. As a result, additional capacity from full fine-tuning does not translate into substantial improvements.
>
> We will include a brief summary of these experiments in the supplementary material.
>
> **Q2**. The adaptation module is relatively small. For example in comparison with LPIPS, which relies on a deep convolutional backbone (e.g., AlexNet or VGG) with tens to hundreds of millions of parameters, while the proposed adaptation module is a small MLP with on the order of $10^4$ parameters. As a result, the adaptation operates as a lightweight post-processing step on top of the LPIPS output. However if we evaluate it for thousands of configurations it requires comparable runtime. We will do runtime measurements experiments and include them to the supplementary.
>
> **Q3**. Yes, our observations are that HDR and SDR gains are simmilar,  we plan to include the analysis of the question to the supplementary.
>
> **Q4**. We have investigated such dependencies; however, at this stage, we have not identified a clear set of characteristics that consistently distinguish metrics that benefit from adaptation from those that do not. We agree that this is an interesting aspect for the reader, and we have therefore included a discussion of this issue in the revised version.
>
> **Q5**. First, all participants provided informed consent prior to participation. They were explicitly notified about the purpose of the study, the type of data being collected, and how this data would be used. Participation was fully voluntary, and users retained the right to withdraw at any stage without any consequences.
>
> Second, we followed strict data minimization principles. No personally identifiable information (PII) such as names, contact details, or precise location data was collected. All responses were stored in an anonymized form, and each participant was represented only by a randomly assigned identifier. This is consistent with app market requirements that user data must be limited to what is strictly necessary and processed in a de-identified manner whenever possible.
>
> Third, regarding compensation, participants were rewarded through crowdsourcing platform-native mechanisms that do not expose personal financial or identity information to the researchers. This ensures that compensation does not introduce additional privacy risks. We also will add links to the user agreement of the platforms.
>
> We will expand Appendix C.2 to explicitly include these details to improve clarity and transparency.
>
>
>
> We also thank you for the suggestion to move the limitations section from the supplementary to the main paper, and we will do so in the revised version. If your concerns have been satisfactorily addressed, we would greatly appreciate your consideration when updating your evaluation of our submission.

---

> > ### Author Rebuttal · Reviewer_Hr6s · 2026-04-03
> >
> > The rebuttal partially addresses the concerns, as it provides clear and technically grounded responses to all questions (e.g., modest gains from fine-tuning, lightweight adaptation, and ethical safeguards), but several answers remain preliminary—relying on hypotheses, lacking deeper analysis (e.g., metric behavior), and deferring important evidence (runtime, HDR/SDR, limitations) to supplementary material rather than fully resolving them in the main paper.

---

> > > ### Author Response · Authors · 2026-04-08
> > >
> > > We thank the reviewer for the thoughtful and constructive comments. While the suggested directions are indeed interesting and valuable, they extend beyond the scope of the present study. That said, the feedback has been helpfull in improving the clarity and overall quality of the manuscript.

---

### Official Review · Reviewer_a64o · 2026-03-13

**Soundness:** 2
**Presentation:** 2
**Significance:** 2
**Originality:** 2
**Overall Recommendation:** 3
**Confidence:** 4

**Summary:**

This paper studies video quality assessment under mobile-device and viewing-condition settings. The main contribution is a new large-scale crowdsourced dataset collected on more than 300 Android devices, with metadata including screen size, display type, brightness, ambient illumination, and HDR/SDR-related properties. The paper also proposes a condition-aware score aggregation approach based on a Blade-Chest formulation and an adaptation module that adjusts existing VQA metrics to specific viewing conditions. Experimental results show that incorporating device and context information can improve Kendall rank correlation over a broad range of full-reference and no-reference quality metrics.

**Compliance With Llm Reviewing Policy:**

Affirmed.

**Key Questions For Authors:**

See weaknesses

**Limitations:**

yes

**Strengths And Weaknesses:**

Strengths:

1. The paper addresses a real and underexplored problem. Many VQA models are developed and benchmarked under limited or laboratory-like conditions, whereas actual mobile viewing occurs under highly diverse conditions. The focus on mobile devices and real-world viewing variability is meaningful.

2. The collection of a large-scale dataset with over 250K annotations and associated hardware/environmental metadata is a substantial contribution.


Weaknesses:

1.  The methodological contribution is incremental. The adaptation module is essentially an MLP conditioned on metric predictions and viewing variables, and the “condition pool” idea is not yet theoretically grounded. The paper does not fully establish why  Blade-Chest-based aggregation is preferable to simpler or more standard alternatives for conditional preference aggregation.

2. The paper focuses on viewing environments, but crowd-sourcing is hard to control. It is very difficult to guarantee that a user's environment stays exactly the same during the test.

3. The writing quality and clarity of this manuscript need to be improved.

---

> ### Author Rebuttal · Authors · 2026-03-31
>
> Thank you for your careful review and the time you have invested in evaluating our work. We sincerely appreciate your efforts. Below, we address your concerns.
>
> **W1**. We would like to clarify that the primary goal of the proposed framework is not to introduce a novel architecture, but to demonstrate a principled way to incorporate viewing conditions into VQA/IQA pipelines. In this context, we intentionally use a simple MLP as the adaptation module, serving as a minimal and stable function approximator to validate the concept. A detailed architectural comparison or ablation over alternative models is beyond the scope of this work.
>
> Regarding the condition pool, we agree that its current description is not sufficiently detailed, and we expanded this part in the revised version. The condition pool is a key component **motivated by the inherent sparsity of subjective data collected under heterogeneous viewing conditions**. In the first stage, the Blade–Chest aggregation operates on sparse pairwise comparisons collected under varying conditions to estimate latent subjective scores and learn condition-aware transformations. Once these scores are obtained, the original sparse observations are no longer sufficient for robust training of downstream models. The condition pool addresses this by **enabling sampling of viewing conditions independently from the original annotations**, effectively augmenting the training process and improving generalization, including to conditions not explicitly observed in the dataset.
> Importantly, this mechanism is only feasible due to the use of the Blade–Chest model, which explicitly incorporates viewing conditions into the aggregation process.
>
> We further note that the preference for Blade--Chest aggregation over a broad class of pairwise aggregation methods can be theoretically motivated by introducing a general matchup function. Let $M(x_i, x_j)$ denote a matchup function. This function can be defined in a general form, provided that the following conditions hold:
>
> * $M(x_i, x_j) \in \mathbb{R}$; positive values indicate that item $i$ is more likely to win, while negative values indicate the opposite. When $M(x_i, x_j) = 0$, both outcomes are equally likely;
> * as $M(x_i, x_j) \to +\infty$, $P(x_i \succ x_j) \to 1$, and vice versa;
> *  $M(x_i, x_j) = -M(x_j, x_i)$, ensuring consistency of pairwise probabilities.
>
>
> Within this framework, classical models can be expressed as specific choices of $M$. For example, the Bradley--Terry model corresponds to
> $$
> M(x_i, x_j) = s_i - s_j,
> $$
> where $s_i$ are latent scores.
>
> A key limitation of these formulations is that the matchup function depends only on item-specific representations and their differences. As a result, they inherently assume that preferences are invariant across external conditions, and do not provide a principled mechanism to incorporate viewing conditions $z$ into the comparison in a way that modifies the interaction between items. Therefore, in these models, when $z$ is treated as an independent feature and presented in the same form for each pairwise comparison, its contribution cancels out.
>
> While it is possible to directly learn such dependencies using ML models, in practice this approach is not robust due to the sparsity of observations across viewing conditions. We experimentally evaluated such direct learning strategies and observed no improvement, which we attribute to insufficient coverage of the condition space. We will include these results in the supplementary material.
>
>
> **W2**. We fully agree that accurate control in crowd-sourced studies is crucial. To mitigate fluctuations in viewing conditions during the test, we continuously logged key dynamic parameters at a one-second resolution during the test:
>
> *Supp. C.2:
> ...the application logged limited device and environmental metadata required for the analysis of viewing conditions, including device model and basic hardware specifications, as well as screen brightness and ambient light levels sampled once per second during the task...*
>
> Furthermore, to ensure data reliability, we explicitly filtered out unstable measurements:
>
> *Supp. C.3:
> ...we removed measurements with abrupt and physically implausible changes in illumination, such as large spikes or drops occurring within one-second intervals, as these are often caused by sensor noise, temporary occlusions, or background system events...*
>
> We also plan to release not only aggregated statistics, but the full time-resolved logs collected during each test session. This will enable future work to apply alternative filtering strategies. Thank you for the valuable suggestion!
>
> **W3**. The paper has been revised by a native English-speaking reviewer. Additionally, following your comment, we conducted an extra pass and addressed several minor clarity issues throughout the manuscript. Thank you!
>
> If you have no further concerns we would be sincerely grateful if you could consider raising the rating of our submission.

---

> > ### Author Rebuttal · Reviewer_a64o · 2026-04-03
> >
> > Thanks for the rebuttal. The expanded discussion on the matchup function and condition pool improves clarity, but the theoretical motivation for Blade–Chest remains informal. The crowdsourcing safeguards are reasonable yet do not fully address my concern, a robustness analysis under perturbed conditions would be more convincing. Therefore, I will keep my rating.

---

> > > ### Author Response · Authors · 2026-04-08
> > >
> > > Thank you for the continued discussion and for raising these important points. We appreciate the opportunity to clarify our motivation and strengthen the presentation.
> > > To better explain the advantage of the proposed Blade–Chest formulation, we distinguish between two possible strategies for learning from the collected data:
> > > * Score aggregation, followed by their modeling
> > > * Direct learning of pairwise preferences.
> > >
> > > For the aggregation-based approach, the Blade–Chest model is specifically designed to address the limitations of standard formulations in the presence of varying match conditions. As discussed in prior work [1,2], conventional models fail to properly account for such conditional effects within the aggregation stage. While the current formulation already follows established principles, we will further expand the supplementary material with a more detailed formal treatment for additional clarity.
> > >
> > > For direct pairwise learning, our key observation is that the data is inherently sparse in the joint space of content and viewing conditions. As a result, learning a stable mapping that captures the effect of conditions becomes difficult, and generalization to unseen conditions is particularly poor. To support this claim, we conducted an additional experiment (to be included in the revision), where we benchmark Linear models, MLPs, and GAMs as representative approximators classes for direct pairwise preference prediction. The results show that none of these approaches achieves consistent improvements, indicating that direct learning is insufficient in this setting. This highlights the necessity of the Blade–Chest framework combined with the condition pool.
> > >
> > > Metric  | Gain for Linear  |  Gain for MLP     | Gain for GAM
> > > --------|---------|---------|---------
> > > SSIM    | -0.0043 |  0.0067 |  0.0021
> > > VMAF    |  0.0018 |  0.0074 | -0.0036
> > > LPIPS   | -0.0065 |  0.0059 |  0.0007
> > > DISTS   |  0.0032 | -0.0028 |  0.0080
> > > DOVER   | -0.0087 |  0.0046 |  0.0013
> > >
> > >
> > > As an additional validation of robustness, we performed an experiment with controlled perturbations of key variables (screen brightness and ambient illumination). Specifically, we applied ±10% noise to these parameters for 10%, 20%, and 50% of users, and repeated the full training pipeline. The results demonstrate that the proposed method remains stable under such perturbations, with only marginal degradation compared to the clean setting.
> > >
> > > Metric  | Clean  | 10% users | 20% users | 50% users
> > > --------|--------|-----------|-----------|-----------
> > > SSIM    | -0.001 | -0.001    | +0.003    | -0.001
> > > VMAF    |  0.163 |  0.151    |  0.131    |  0.074
> > > LPIPS   |  0.229 |  0.199    |  0.175    |  0.099
> > > DISTS   |  0.171 |  0.138    |  0.106    |  0.019
> > > DOVER   |  0.109 |  0.085    |  0.071    |  0.047
> > >
> > >
> > > We emphasize that the primary contribution of this work lies in (i) introducing a novel problem formulation for condition-aware quality assessment, (ii) collecting a large-scale dataset with rich viewing condition metadata, and (iii) proposing a practical and effective framework that enables learning under these conditions.
> > >
> > > In light of the clarifications and additional experiments, we kindly ask you to reconsider your rating, as we believe the work provides a meaningful contribution to the field of VQA.
> > >
> > > [1] Chen, S., & Joachims, T. (2016, February). Modeling intransitivity in matchup and comparison data. In Proceedings of the ninth acm international conference on web search and data mining (pp. 227-236).
> > > [2] Makhijani, R., & Ugander, J. (2019, May). Parametric models for intransitivity in pairwise rankings. In The World Wide Web Conference (pp. 3056-3062).

---

### Decision · Program_Chairs · 2026-04-30

**Decision:**

Accept (regular)

**Comment:**

This paper tackles a highly relevant and historically under-explored problem in Video Quality Assessment (VQA): how diverse mobile device characteristics and ambient viewing conditions impact perceived video quality. The authors introduce an impressive, large-scale dataset comprising over 250,000 crowd-sourced pairwise preferences across more than 300 Android devices, complete with rich environmental metadata such as screen brightness and ambient lighting. To utilize this data, the authors propose a Blade-Chest-based aggregation method and a lightweight adaptation module designed to adjust existing VQA metric predictions based on these specific viewing conditions. The reviewing committee unanimously praised the dataset's unprecedented scale and originality, agreeing that bridging the gap between controlled laboratory VQA and in-the-wild, real-world media consumption is a highly valuable contribution to both academic research and industrial streaming applications.

During the review process, however, reviewers raised several valid critiques regarding the methodological depth of the proposed adaptation framework. Primary concerns centered on the simplicity of the post-hoc multi-layer perceptron (MLP) adaptation module, the lack of explicit content-condition interaction modeling, and questions regarding the statistical reliability of crowd-sourced data under such heterogeneous conditions. In their rebuttal, the authors provided extensive additional analyses, including comparisons against simpler baselines (such as Linear and GAM models) to justify the Blade-Chest formulation, robustness tests under perturbed conditions, and supplementary experiments incorporating spatial and temporal content descriptors. While some reviewers maintained their reservations—noting that the method still treats conditions somewhat as auxiliary features and that key analyses were deferred to supplementary materials—the general consensus is that the sheer utility of the released dataset and the effectively demonstrated proof-of-concept baseline outweigh these methodological limitations. Consequently, I recommend this paper for a Weak Accept, as it introduces a vital benchmark and a solid foundational framework that the community can meaningfully build upon.